# Look, Listen, and Answer: Overcoming Biases for Audio-Visual Question Answering

**Jie Ma**[†1*], **Min Hu**[†1,2], **Pinghui Wang**[1], **Wangchun Sun**[1], **Lingyun Song**[4],
**Hongbin Pei**[1], **Jun Liu**[3], **Youtian Du**[1]

[1] MOE KLINNS Lab, Xi'an Jiaotong University
[2] China Mobile System Integration Co.
[3] School of Computer Science and Technology, Xi'an Jiaotong University
[4] School of Computer Science, Northwestern Polytechnical University
[†] Equal Contribution
[*] Corresponding Author

## Abstract

Audio-Visual Question Answering (AVQA) is a complex multi-modal reasoning task, demanding intelligent systems to accurately respond to natural language queries based on audio-video input pairs. Nevertheless, prevalent AVQA approaches are prone to overlearning dataset biases, resulting in poor robustness. Furthermore, current datasets may not provide a precise diagnostic for these methods. To tackle these challenges, firstly, we propose a novel dataset, *MUSIC-AVQA-R*, crafted in two steps: rephrasing questions within the test split of a public dataset (*MUSIC-AVQA*) and subsequently introducing distribution shifts to split questions. The former leads to a large, diverse test space, while the latter results in a comprehensive robustness evaluation on rare, frequent, and overall questions. Secondly, we propose a robust architecture that utilizes a multifaceted cycle collaborative debiasing strategy to overcome bias learning. Experimental results show that this architecture achieves state-of-the-art performance on MUSIC-AVQA-R, notably obtaining a significant improvement of 9.32%. Extensive ablation experiments are conducted on the two datasets mentioned to analyze the component effectiveness within the debiasing strategy. Additionally, we highlight the limited robustness of existing multi-modal QA methods through the evaluation on our dataset. We also conduct experiments combining various baselines with our proposed strategy on two datasets to verify its plug-and-play capability. Our dataset and code are available at `https://github.com/reml-group/MUSIC-AVQA-R`.

## 1 Introduction

Humans possess the extraordinary capacity to seamlessly integrate auditory and visual cues, effectively establishing a cohesive relationship between visual and auditory stimuli [1]. Audio-Visual Question Answering (AVQA) [2–5, 3] seeks to enable intelligent systems to acquire this capability and produce answers based on provided natural language questions. It requires the system to learn high-order interaction representations of the concepts encompassed with audio, video, and language modalities. As is known to us [6–8], the high-level reasoning ability of the system mainly relies on large-scale data that does not contain harmful biases or statistical regularities.

Nevertheless, completely avoiding the negative bias in datasets seems challenging. Previous studies [9–12] in visual and extractive QA have investigated the bias from the perspective of changing answer distributions and human-in-the-loop adversarial attacks. Drawing inspiration from these works,

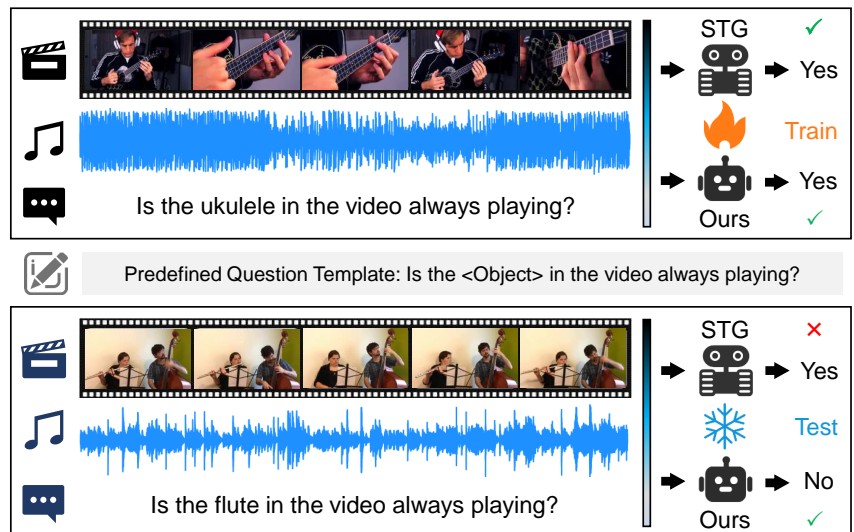

Figure 1: The question in current AVQA datasets is generated by a limited set of predefined templates, which may not be in line with the real-world scenario. Our findings indicate that existing methods [5, 1] such as STG [4] are not robust, which may be attributed to excessive bias learning, such as memorizing statistical regularities between critical question words and answers.

several open questions are proposed for the AVQA task, concerning model evaluations and model designs.

**Question 1: have existing datasets comprehensively measured model robustness?** The questions in the current AVQA dataset [13, 5, 14, 3] are generated by a limited set of predefined templates, such as the 33 templates in the MUSIC-AVQA dataset [4]. Fig. 1 shows the samples in the training and test split, which are produced using a predefined template. The observed difference mainly stems from a single word, leading to a limited vocabulary size of only 93 words. This has the potential to deviate from real-world scenarios. Moreover, current datasets cannot reflect the performance on rare or less common samples, which is an important indicator for evaluating model robustness [15, 16].

**Question 2: have existing methods overcome the data bias?** We found that existing methods [5, 1, 17, 18] such as STG [4] are brittle for the question with rare answers. This may be attributed to memorizing the statistical regularity between critical question words and answers, such as the connection between "Is", "Playing", and "Yes". Specifically, the experimental result [4] shows that STG achieves an accuracy of 54.09% on the test split of MUSIC-AVQA only given questions.

In this paper, we present the development of a novel dataset called MUSIC-AVQA-R, which aims to address the first question precisely. The dataset complements MUSIC-AVQA [4] and provides a more refined diagnostic for current AVQA methods. *To preserve the inherent bias, we maintain the original training and validation splits of the MUSIC-AVQA dataset.* In contrast, we employ a human-machine collaboration mechanism to rephrase the question in the test split. This ensures diverse and natural question forms while remarkably expanding the number of questions from **9,129** to **211,572**. We introduce a distribution shift based on answer distributions of specific question types. This allows us to measure performance on both frequent (in-distribution) and rare (out-of-distribution) data simultaneously.

To tackle the second question, we propose a robust framework that applies a *Multifaceted Cycle Collaborative Debiasing* (MCCD) strategy. Specifically, the strategy introduces a novel optimization objective, which enlarges the distribution difference between uni-modal (question, audio, and video) and multi-modal logit. By doing so, our model becomes less prone to learning biases from individual modalities. Intuitively, we cannot choose the correct answer based on only one modality. Hence, MCCD employs cycle guidance to constrain the logit distribution of each modality, thereby promoting the similarity of uni-modal logit distribution. The experimental results demonstrate that our framework yields significant improvements on both datasets, with a particularly notable enhancement of 9.32% observed on the MUSIC-AVQA-R dataset.

In summary, our contributions are fourfold: (1) We propose a novel dataset MUSIC-AVQA-R and a set of respective evaluation metrics. This enables us to thoroughly evaluate the reasoning behavior of AVQA models and characterize their generalization capabilities in in- and out-of-distribution scenarios. (2) We present an AVQA architecture that incorporates the MCCD strategy to overcome training biases. To the best of our knowledge, this is the first work to systematically explore biases in the AVQA task from model evaluations as well as model designs. (3) We conduct extensive experiments on MUSIC-AVQA and MUSIC-AVQA-R to verify the effectiveness and superiority of our proposed architecture and debiasing strategy. (4) We evaluate 13 recent multimodal QA methods on the proposed dataset and show their limited ability to generalize not only in-distribution scenarios but also in out-of-distribution situations.

## 2 Related Work

### 2.1 Model Robustness Evaluation

Despite the notable achievements of QA datasets [19–23, 3], they suffer from biases, resulting in incomplete evaluations. In recent years, numerous studies have tackled this issue from various perspectives [24–27].

One avenue of research [12, 28, 29] reorganizes existing datasets, thereby making the distribution between training and testing splits significantly different or even reversed. The reorganized datasets reflect the performance in the out-of-distribution situation but lack measurement in the in-distribution scenario. To this end, GQA-OOD [30] introduces the distribution shift in both the validation and test splits to assess visual QA models in both scenarios simultaneously. Nevertheless, the number of questions in the GQA-OOD test split is only 2,796, which may not reflect the real generalization ability of visual QA models due to the presence of a limited number of testing samples [31]. Inspired by the adversarial attack, another line of works [32, 33] regard the dataset construction as a game played by two parties: a human annotator and a well-trained model. Only samples generated by humans that successfully attack the model are incorporated into the dataset. In addition, there exists another line of work [14] that complements videos and questions to obtain balanced training data.

Different from the mentioned works, our dataset, MUSIC-AVQA-R, not only prioritizes question diversity but also considers the volume of test samples. This enhances the precision and comprehensiveness of robustness evaluation. Moreover, we recognize the formidable challenge of obtaining completely pure training data. As such, we opt to retain the inherent bias present in both the training and validation splits. Our primary objective is to inspire the community to enhance model robustness through the implementation of debiasing strategies, rather than striving for balanced training data. *Remarkably, to the best of our knowledge, our dataset is the first AVQA dataset explicitly designed for robustness evaluation.*

### 2.2 Bias Dependency Elimination

A variety of debiasing QA methods [34–37] have been proposed to overcome bias memorization. These methods can be divided into four classes [24]: ensemble learning, data augmentation, contrastive learning, and answer re-ranking.

Ensemble learning methods [38, 28, 39, 6, 40, 41] typically leverage a combination of a bias learner and a vanilla QA model to comprehensively predict answers. Data augmentation methods [42–45] generate additional question-answer pairs to balance the data distribution. Based on the positive and negative sample generation, contrastive learning-based methods [46–48] strive to learn an embedding space where similar sample pairs are closely clustered while disparate ones are distinctly separated. Consequently, the vanilla QA method is optimized jointly through contrastive and QA losses. Answer re-ranking methods [34, 49–52] primarily focus on reordering the answers predicted by the vanilla QA model to enhance context comprehension, such as vision grounding.

To the best of our knowledge, COCA [53] is the only work to mitigate the bias learning in the AVQA task, which first employs causal regularization to intervene bias-irrelevant causal effects and then introspects predictions. Unlike the mentioned works, which only consider language biases, our method considers audio, vision, language biases, and their collaboration. *The proposed MCCD strategy features plug-and-play capability, enhancing the debiasing potential of baseline methods.*

## 3 Dataset Creation and Analysis

We introduce the first dataset, MUSIC-AVQA-R, to evaluate the robustness of AVQA models. The construction of this dataset involves two key processes: rephrasing and splitting. The former involves the rephrasing of questions in the test split of MUSIC-AVQA, and the latter is dedicated to the categorization of questions into frequent (head) and rare (tail) subsets.

### 3.1 Rephrasing

The questions within the existing dataset [5, 4] are formulated using a restricted collection of pre-defined templates. To augment diversity and reality, we employ a rephrasing tool[1] to rephrase each question 25 times. To ensure the rephrasing quality, three annotators participate in a verification process where their consensus through voting is required. They are all senior students in the field of information science, with one specializing in computer science and the other two in automation. Their extensive professional background equips them with the ability to assess whether the above rephrasing fulfills the requirement. Rephrasings are incorporated into the dataset only when two or more individuals validate the quality of the modifications. According to the statistics, 92.4% of rephrasings pass this validation, and the Fleiss Kappa value used to measure vote consistency is 0.839. Please see details in Table 4 of Appendix B. These results strongly suggest an exceptionally high quality of the rephrasing efforts. Fig. 2 illustrates the distribution of rephrased questions based on their initial three words. We see that our rephrasing questions have various formats, and the comparison between the two datasets is shown in Fig. 6 of Appendix B. The vocabulary size of our dataset is **465**, which is **5x** larger than MUSIC-AVQA. These results indicate that our dataset is more in line with the real-world scenario. Furthermore, an expansion of the question count within the test split has been implemented, escalating from **9,129** to **211,572** questions. This substantial increase in the volume of test samples enhances the precision of evaluations for AVQA methods.

### 3.2 Splitting

To provide a precise diagnostic for AVQA models, we introduce a distribution shift based on answer distributions of specific question types, following [30]. Guided by this distribution, we categorize rephrased questions into *head* and *tail*, enabling the assessment of in-distribution and out-of-distribution performance, respectively. We also utilize the *overall* performance to assess the model effectiveness on the entire test split.

Specifically, to characterize the distribution shift, we first utilize the annotation for question types, including "Existential", "Location", "Counting", "Comparative", and "Temporal", to group questions. Fig. 3(a) illustrates the answer distribution of the "Temporal" questions within the AVQA task. We see that the answer presents a long-tailed distribution. The distribution of other types is given in Appendix B. It is essential to note that MUSIC-AVQA encompasses three tasks: audio QA, visual QA, and AVQA.

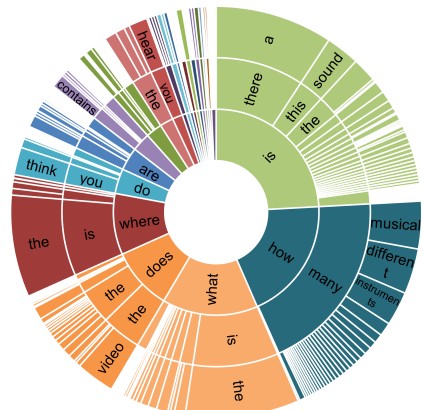

Figure 2: Distribution of rephrasing questions based on the first three words.

Next, we characterize the answer balance using Shannon entropy, expressed as $H(A) = -\sum_{i=1}^{N} p(a_i) \log p(a_i)$, where $H(A)$ is the entropy of an answer set $A$ for a certain question type, $N$ is the number of answer classes, and $p(a_i)$ represents the probability of answer class $i$. It is important to note that the entropy depends on the number of answer classes, which exhibits significant variability across different question groups. To facilitate meaningful comparisons, we normalize $H(A)$ of each group by $\log(N)$: $\bar{H}(A) = \frac{H(A)}{\log(N)}$, with $\log(N)$ representing the entropy of a uniform distribution of size $N$. Refer to Appendix C for detailed proof. Thus, the normalized entropy $\bar{H}(A)$ indicates the proximity of the distribution $H(A)$ to a

[1] https://quillbot.com/paraphrasing-tool

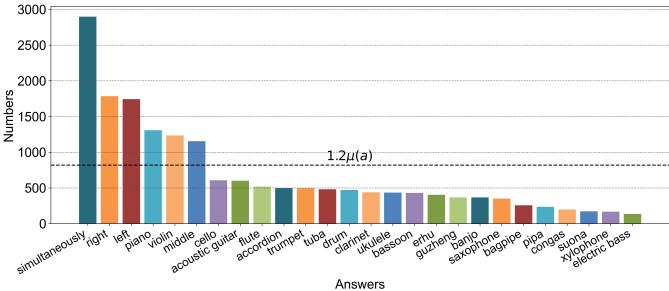
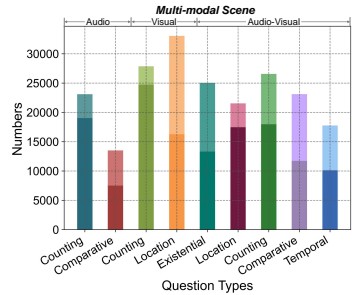

(a) Answer distributions of "Temporal" questions in the AVQA task.    (b) Statistics of head and tail samples.

Figure 3: Statistics visualization for MUSIC-AVQA-R. $\mu(a)$ is the average number of answers in a group. The dark color on the right denotes the number of head samples, while the light-colored area denotes that of tail samples.

uniform distribution. We preserve the group with a normalized entropy below a threshold of 0.9, which aims at selecting imbalanced groups.

Finally, we categorize the samples into *head* and *tail* classes. We define the *tail* class as class $i$ with $|a_i| \leq 1.2\mu(a)$ following [30], where $|a_i|$ represents the number of samples belonging to answer class $i$, and $\mu(a)$ denotes the average sample count for a group. Consequently, the *tail* samples are rare, while the *head* samples are more prevalent within a group. Fig. 3(b) illustrates the statistics of head and tail samples across various groups within each task.

## 4  Method

To mitigate bias learning, we propose a robust AVQA architecture that integrates a multifaceted cycle collaborative debiasing strategy. Fig. 4 illustrates an overview of our proposed architecture. It first learns the uni-modal and multimodal representations using a pre-trained model. Then, the architecture utilizes distinct bias learners to capture uni-modal biases. Finally, a collaborative debiasing strategy is leveraged to magnify the disparity between fusion logit and bias logit, obtained based on multi-modality and uni-modality representations, respectively. Meanwhile, a cycle guidance mechanism is employed to maintain the similarity between bias logit. The aforementioned procedure is only carried out in the test split of MUSIC-AVQA. Consequently, our proposed dataset, MUSIC-AVQA-R, allows for a more precise and comprehensive evaluation of models handling data biases.

**Uni-modal Embedding.** Given an AVQA sample comprising a video sequence and a corresponding question, we initially partition the sequence, consisting of visual and audio tracks, into $T$ non-overlapping pairs of visual and audio segments, denoted as $\{V_t, A_t\}_{t=1}^{T}$, where each segment spans one second. Subsequently, a distinct embedding layer is employed to acquire uni-modal embeddings. Specifically, we employ a pre-trained VGGish model [54] with fixed parameters, which is a VGG-like audio processing network, to obtain an audio embedding vector. For video embedding vectors, we employ a pre-trained ResNet-18 with fixed parameters on the frames. The VisualBert model [55] is applied to obtain a word-level question embedding vector. To ensure dimension matching, distinct linear layers are applied to the aforementioned vectors, resulting in uni-modal embeddings $\mathbf{A}_i^e, \mathbf{V}_i^e \in \mathbb{R}^{T \times 768}$ and $\mathbf{Q}_i^e \in \mathbb{R}^{l \times 768}$, where $l$ is the question length.

**Uni- and Multi-modal Representation.** We leverage VisualBert to obtain both uni-modal and multi-modal representations, represented as $\mathbf{A}_i^c, \mathbf{V}_i^c, \mathbf{Q}_i^c, \mathbf{M}_i^c \in \mathbb{R}^{768}$. In the case of uni-modal learning, we exclusively input the aforementioned uni-modal embeddings to VisualBert. For multi-modal learning, we treat question embeddings as queries, concatenate video and audio embeddings as context, and leverage VisualBert to perform multi-modal interaction. Then, we apply a linear projection on the representation to obtain the multi-modality logit $\hat{\mathbf{y}}_i^m \in \mathbb{R}^{42}$, where 42 denotes the number of possible answers.

**Uni-modal Bias Learning.** AVQA may involve various harmful uni-modal biases, encompassing biases associated with audio, video, and language, respectively. To capture these uni-modal biases,

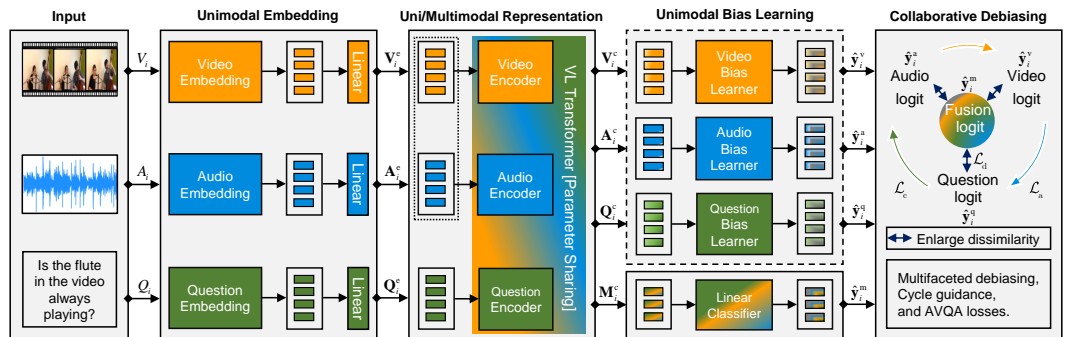

Figure 4: Robust AVQA architecture to overcome bias learning. Our MCCD strategy is plug-and-play, allowing seamless integration with other AVQA methods.

we utilize a bias learner that takes only one of the three modalities as input. Specifically, distinct non-linear multi-layer perceptron layers serve as the learners, producing the corresponding logit $\hat{\mathbf{y}}_i^{\mathrm{a}}, \hat{\mathbf{y}}_i^{\mathrm{v}}, \hat{\mathbf{y}}_i^{\mathrm{q}} \in \mathbb{R}^{42}$ on the answer space. It should be noted that these bias learners are removed during the testing stage.

**Collaborative Debiasing.** To eliminate bias learning, we propose a *multifaceted cycle collaborative debiasing* (MCCD) strategy. It first reduces the bias impact from multiple views by enlarging the dissimilarity between uni-modal and multi-modal logit. This discrepancy enlargement $\mathcal{L}_{\mathrm{d}}$ is implemented by the joint inverse distance:

$$\mathcal{L}_{\mathrm{d}} = \frac{\alpha}{3K} \sum_{i=1}^{K} \left( \frac{1}{d_i^{\mathrm{a}}} + \frac{1}{d_i^{\mathrm{v}}} + \frac{1}{d_i^{\mathrm{q}}} \right), \tag{1}$$

where $K$ is the batch size, $\alpha$ is used to balance optimization, $d_i^{\mathrm{a}}$ denotes the Euclidean distance between audio logit and multi-modality logit, $d_i^{\mathrm{v}}$ represents the distance between video logit and multi-modality logit, $d_i^{\mathrm{q}}$ is the distance between question logit and multi-modality logit, and $\epsilon = 1e{-}5$ is added to the denominator to avoid division by zero.

Intuitively, relying solely on one modality for answer prediction may result in similar logit distributions. Therefore, MCCD employs cycle guidance to constrain the distribution of uni-modal logit. This guidance $\mathcal{L}_{\mathrm{c}}$ is implemented by the Kullback–Leibler divergence:

$$\mathcal{L}_{\mathrm{c}} = \frac{\beta}{3} \left( \mathcal{L}_{\mathrm{qa}} + \mathcal{L}_{\mathrm{av}} + \mathcal{L}_{\mathrm{vq}} \right), \tag{2}$$

where $\beta$ is the factor to control weight, $\mathcal{L}_{\mathrm{qa}} = \frac{1}{K} \sum_{i=1}^{K} \hat{\mathbf{y}}_i^{\mathrm{q}} \left( \log \hat{\mathbf{y}}_i^{\mathrm{q}} - \log \hat{\mathbf{y}}_i^{\mathrm{a}} \right)$ denotes the relative entropy between the question $\hat{\mathbf{y}}_i^{\mathrm{q}}$ and audio logit $\hat{\mathbf{y}}_i^{\mathrm{a}}$, $\mathcal{L}_{\mathrm{av}} = \frac{1}{K} \sum_{i=1}^{K} \hat{\mathbf{y}}_i^{\mathrm{a}} \left( \log \hat{\mathbf{y}}_i^{\mathrm{a}} - \log \hat{\mathbf{y}}_i^{\mathrm{v}} \right)$ is the relative entropy between the audio $\hat{\mathbf{y}}_i^{\mathrm{a}}$ and video logit $\hat{\mathbf{y}}_i^{\mathrm{v}}$, and $\mathcal{L}_{\mathrm{vq}} = \frac{1}{K} \sum_{i=1}^{K} \hat{\mathbf{y}}_i^{\mathrm{v}} \left( \log \hat{\mathbf{y}}_i^{\mathrm{v}} - \log \hat{\mathbf{y}}_i^{\mathrm{q}} \right)$ represents the relative entropy between the video $\hat{\mathbf{y}}_i^{\mathrm{v}}$ and question logit $\hat{\mathbf{y}}_i^{\mathrm{q}}$.

Finally, we utilize the summation of $\mathcal{L}_{\mathrm{d}}$, $\mathcal{L}_{\mathrm{c}}$ and $\mathcal{L}_{\mathrm{a}}$ to optimize the parameters of our method. $\mathcal{L}_{\mathrm{a}} = -\frac{1}{K} \sum_{i=1}^{K} \mathbf{y}_i^{\mathrm{f}} \log \hat{\mathbf{y}}_i^{\mathrm{f}}$ is the loss of answer prediction that is regarded as a multi-classification problem, where $\mathbf{y}_i^{\mathrm{f}}, \hat{\mathbf{y}}_i^{\mathrm{f}}$ denote the one-hot answer label and logit of multi-modality fusion, respectively. The training details are shown in Appendix A.

## 5 Experiments

### 5.1 Dataset and Evaluation

MUSIC-AVQA [4], which contains training, validation, and testing splits with 31,927, 4,568, and 9,129 QA pairs, is developed by gathering questions for 9,288 musical performances. The questions are produced by a limited set of pre-defined templates. The videos, sourced from YouTube, include solo performances, ensembles of the same instruments, and ensembles of different instruments. This dataset consists of three tasks: audio QA, visual QA, and AVQA. The standard accuracy is used to

evaluate model performance on the mentioned tasks. To comprehensively assess model robustness, we conduct rephrasing and splitting on the test split, expanding the question count from 9,129 to 211,572. Owing to the introduction of distribution shift, our proposed dataset provides three metrics: *head accuracy, tail accuracy, and overall accuracy*, to evaluate models precisely. The test split comparison between MUSIC-AVQA and MUSIC-AVQA-R is shown in Appendix D.1. We can see that the latter exhibits a larger test sample space.

Table 1: Experimental results (%) on the MUSIC-AVQA test split. EXIST, LOC, CNT, COMP, and TEMP, which are question types, denote "Existential", "Location", "Counting", "Comparative", and "Temporal", respectively. Avg. denotes the average accuracy.

| Method | MCCD | Audio QA | | | Visual QA | | | AVQA | | | | | | All |
|---|---|---|---|---|---|---|---|---|---|---|---|---|---|---|
| | | CNT | COMP | Avg. | CNT | LOC | Avg. | EXIST | LOC | CNT | COMP | TEMP | Avg. | Avg. |
| FCNLSTM | × | 70.45 | 66.22 | 68.88 | 63.89 | 46.74 | 55.21 | 82.01 | 46.28 | 59.34 | 62.15 | 47.33 | 60.06 | 60.34 |
| | ✓ | 70.99 | 66.50 | **69.34** | 66.08 | 59.02 | **62.51** | 83.50 | 57.17 | 60.47 | 61.58 | 57.54 | **64.11** | **64.61** |
| CONVLSTM | × | 74.07 | 68.89 | 72.15 | 67.47 | 54.56 | 60.94 | 82.91 | 50.81 | 63.03 | 60.27 | 51.58 | 62.24 | 63.65 |
| | ✓ | 72.76 | 69.53 | 71.57 | 69.59 | 58.12 | **63.79** | 82.69 | 56.09 | 62.13 | 62.03 | 55.11 | **63.87** | **65.21** |
| BiLSTM Attn | × | 70.35 | 47.92 | 62.05 | 64.64 | 64.33 | 64.48 | 78.39 | 45.85 | 56.91 | 53.09 | 49.76 | 57.10 | 59.92 |
| | ✓ | 68.24 | 54.88 | **63.31** | 61.65 | 55.92 | 58.75 | 79.15 | 41.96 | 55.02 | 49.41 | 49.15 | 55.18 | 57.56 |
| HCAttn | × | 70.25 | 54.91 | 64.57 | 64.05 | 66.37 | 65.22 | 79.10 | 49.51 | 59.97 | 55.25 | 56.43 | 60.19 | 62.30 |
| | ✓ | 69.52 | 53.37 | 63.56 | 63.99 | 65.47 | 64.74 | 78.64 | 47.28 | 61.11 | 55.86 | 55.72 | 60.03 | 61.90 |
| MCAN | × | 77.50 | 55.24 | 69.25 | 71.56 | 70.93 | 71.24 | 80.40 | 54.48 | 64.91 | 57.22 | 47.57 | 61.58 | 65.49 |
| | ✓ | 78.27 | 56.57 | **70.27** | 71.93 | 71.18 | **71.55** | 81.48 | 54.24 | 65.77 | 55.86 | 46.84 | 61.54 | **65.74** |
| GRU | × | 72.21 | 66.89 | 70.24 | 67.72 | 70.11 | 68.93 | 81.71 | 59.44 | 62.64 | 61.88 | 60.07 | 65.18 | 67.07 |
| | ✓ | 73.35 | 66.16 | **70.70** | 67.25 | 71.43 | **69.36** | 81.98 | 60.11 | 63.08 | 62.76 | 61.19 | 65.84 | **67.63** |
| HCRN | × | 68.59 | 50.92 | 62.05 | 64.39 | 61.81 | 63.08 | 54.47 | 41.53 | 53.38 | 52.11 | 47.69 | 50.26 | 55.73 |
| | ✓ | 72.17 | 64.65 | **69.40** | 67.42 | 60.82 | **64.08** | 79.66 | 48.70 | 65.14 | 61.22 | 55.72 | **60.40** | **64.20** |
| HME | × | 74.76 | 63.56 | 70.61 | 67.97 | 69.46 | 68.76 | 80.30 | 53.18 | 63.19 | 62.69 | 59.83 | 64.05 | 66.45 |
| | ✓ | 72.96 | 62.29 | 69.03 | 68.76 | 69.31 | **69.03** | 80.77 | 52.61 | 62.92 | 63.03 | 60.71 | **64.19** | 66.33 |
| PSAC | × | 75.64 | 66.06 | 72.09 | 68.64 | 69.79 | 69.22 | 77.59 | 55.02 | 63.42 | 61.17 | 59.47 | 63.52 | 66.54 |
| | ✓ | 75.02 | 65.66 | 71.57 | 69.09 | 69.88 | **69.49** | 79.35 | 53.04 | 61.98 | 61.13 | 57.66 | 62.85 | 66.15 |
| AVSD | × | 72.41 | 61.90 | 68.52 | 67.39 | 74.19 | 70.83 | 81.61 | 58.79 | 63.89 | 61.52 | 61.41 | 65.49 | 67.44 |
| | ✓ | 72.07 | 63.97 | **69.09** | 67.42 | 74.53 | **71.02** | 81.17 | 59.13 | 63.08 | 62.49 | 63.50 | **65.82** | **67.77** |
| LAViT | × | 74.36 | 64.56 | 70.73 | 69.39 | 75.65 | 72.56 | 81.21 | 59.33 | 64.91 | 64.22 | 63.23 | 66.64 | 68.93 |
| | ✓ | 75.12 | 65.49 | **71.57** | 70.43 | 76.73 | **73.62** | 81.38 | 60.33 | 65.30 | 62.49 | 62.29 | 66.42 | **69.24** |
| STG | × | 78.18 | 67.05 | 74.06 | 71.56 | 76.38 | 74.00 | 81.81 | 64.51 | 70.80 | 66.01 | 63.23 | 69.54 | 71.52 |
| COCA | × | 79.35 | 66.50 | 74.61 | 72.35 | 76.08 | 74.24 | 83.50 | 64.02 | 70.99 | 63.40 | 64.48 | 69.47 | 71.64 |
| **Ours** | ✓ | 83.87 | 71.04 | **79.14** | 79.78 | 76.73 | **78.24** | 80.87 | 51.63 | 71.46 | 64.67 | 64.60 | 67.13 | **72.20** |
| LAVisH | × | 81.32 | 63.30 | 74.67 | 79.20 | 80.57 | 79.89 | 83.40 | 65.22 | 72.96 | 64.03 | 66.18 | 70.57 | 73.76 |
| | ✓ | 80.33 | 63.80 | 74.24 | 78.86 | 81.31 | **80.10** | 73.91 | 65.22 | 73.91 | 64.31 | 66.55 | **70.80** | **73.87** |

## 5.2 Implementation Details

During the data pre-processing stage, audio and video are sampled at rates of 16 kHz and 1 fps, respectively. In the uni-modal embedding module, we employ ResNet-18 and VGGish to obtain 512-dimensional and 128-dimensional embeddings of the visual and audio segments, respectively. In the uni-modal bias learning module, the hidden layer size of the bias learner is set to 768. In the collaborative debiasing module, we set the factors $\alpha$ and $\beta$ to $1e$-2 and $3e$-1 for optimization equilibrium, respectively. During the training stage, the initial learning rate is set to $3e$-5, decaying by 0.5 every 20 epochs. The maximum epoch and batch size are set to 150 and 64, respectively. We use the Adam optimizer to train our architecture and save the model that achieves the best performance on the validation split. The experiments for all methods, except LAVisH, are conducted using a single NVIDIA Tesla V100 GPU. The experiment for LAVisH is run on two NVIDIA Tesla A100 GPUs. The other details are shown in Appendix D.2.

## 5.3 Baselines

We select 14 previous state-of-the-art multi-modal QA methods as baselines to verify the effectiveness of the proposed architecture and investigate the robustness of these methods. Audio QA methods: FCNLSTM [56], and CONVLSTM [56]. Visual QA methods: GRU [57], BiLSTM Attn [58], HCAttn [59], and MCAN [60]. Video QA methods: HME [61], PSAC [62], and HCRN [63]. AVQA methods: AVSD [13], LAViT [5], STG [4], COCA [53] and LAVisH [1]. We abstain from reassessing COCA on the MUSIC-AVQA-R dataset due to its lack of publicly available code. The baseline introductions are shown in Appendix D.3. Due to the particularly slow computation speed of STG, we do not conduct experiments with STG+MCCD. Due to computing power limitations, we reevaluate LAVisH with a batch size of 2.

Table 2: Experimental results (%) on the MUSIC-AVQA-R test split. The question types, such as CNT and COMP, are introduced in Table 1. H and T denote the head and tail accuracy. There is no publicly available code for COCA.

| Method | MCCD | Audio QA | | | | Visual QA | | | | AVQA | | | | | | | | | | All |
|---|---|---|---|---|---|---|---|---|---|---|---|---|---|---|---|---|---|---|---|---|
| | | CNT | | COMP | | CNT | | LOC | | EXIST | | LOC | | CNT | | COMP | | TEMP | | Avg. |
| | | H | T | H | T | H | T | H | T | H | T | H | T | H | T | H | T | H | T | |
| FCNLSTM | × | 66.23 | 36.48 | 64.78 | 51.14 | 61.75 | 5.31 | 54.86 | 51.06 | 64.76 | 78.52 | 46.66 | 57.30 | 62.69 | 7.23 | 43.13 | 71.67 | 37.02 | 30.78 | 54.12 |
| | ✓ | 62.51 | 34.44 | 61.19 | 51.26 | 61.11 | 5.66 | 57.73 | 50.36 | 62.48 | 82.40 | 45.49 | 60.09 | 62.07 | 7.16 | 44.55 | 69.46 | 36.55 | 30.74 | **54.55** |
| CONVLSTM | × | 70.22 | 41.14 | 67.50 | 52.93 | 62.11 | 9.17 | 53.44 | 49.88 | 60.08 | 84.82 | 46.46 | 59.90 | 56.52 | 8.18 | 43.29 | 72.52 | 41.54 | 45.12 | 55.20 |
| | ✓ | 68.38 | 41.58 | 68.39 | 52.10 | 61.46 | 9.56 | 54.17 | 50.33 | 59.61 | 83.11 | 55.29 | 56.52 | 59.13 | 7.82 | 45.31 | 72.70 | 41.26 | 45.40 | **55.74** |
| BiLSTM Attn | × | 73.68 | 46.32 | 21.51 | 77.58 | 64.30 | 0.00 | 53.92 | 42.01 | 87.51 | 21.14 | 35.16 | 43.75 | 62.85 | 2.18 | 27.61 | 74.38 | 17.58 | 31.32 | 48.84 |
| | ✓ | 73.30 | 45.16 | 20.71 | 77.48 | 64.41 | 0.00 | 56.08 | 42.54 | 87.47 | 21.04 | 34.47 | 43.51 | 63.33 | 2.18 | 26.01 | 75.48 | 17.92 | 32.67 | **49.55** |
| HCAttn | × | 61.67 | 41.63 | 59.09 | 47.14 | 56.52 | 9.20 | 67.01 | 53.16 | 66.57 | 61.13 | 37.05 | 42.48 | 59.53 | 12.48 | 48.81 | 60.12 | 33.82 | 39.26 | 51.90 |
| | ✓ | 62.50 | 41.43 | 58.89 | 47.42 | 56.65 | 8.85 | 67.31 | 52.92 | 66.82 | 59.87 | 38.25 | 42.53 | 59.38 | 12.42 | 57.39 | 52.01 | 32.84 | 39.55 | **52.29** |
| GRU | × | 66.92 | 48.63 | 58.29 | 59.61 | 64.37 | 11.79 | 57.68 | 57.66 | 76.30 | 64.76 | 41.05 | 45.61 | 60.71 | 18.68 | 57.19 | 57.38 | 31.02 | 40.67 | 55.21 |
| | ✓ | 69.94 | 48.09 | 56.31 | 63.77 | 66.24 | 13.36 | 63.55 | 57.59 | 83.04 | 54.16 | 43.36 | 43.36 | 57.89 | 18.36 | 53.93 | 59.65 | 30.82 | 38.23 | **55.70** |
| MCAN | × | 75.02 | 60.16 | 58.89 | 50.09 | 64.58 | 26.69 | 66.48 | 62.25 | 51.29 | 67.29 | 46.11 | 61.61 | 64.76 | 25.28 | 50.57 | 52.40 | 34.64 | 58.05 | 57.27 |
| | ✓ | 73.53 | 56.14 | 68.31 | 39.44 | 65.51 | 29.40 | 68.41 | 60.09 | 58.80 | 61.90 | 46.75 | 60.61 | 60.54 | 31.89 | 69.09 | 44.94 | 32.44 | 57.78 | **58.22** |
| HCRN | × | 55.53 | 53.31 | 47.17 | 32.44 | 41.87 | 23.55 | 39.40 | 51.27 | 41.81 | 65.45 | 46.62 | 42.72 | 54.58 | 19.57 | 33.33 | 36.87 | 40.47 | 44.13 | 43.92 |
| | ✓ | 51.96 | 49.21 | 43.42 | 36.78 | 41.13 | 20.71 | 37.79 | 50.99 | 44.38 | 58.40 | 35.05 | 46.33 | 54.39 | 20.90 | 34.50 | 33.14 | 40.13 | 44.00 | 42.87 |
| HME | × | 62.60 | 53.95 | 54.97 | 58.29 | 50.95 | 16.46 | 73.25 | 58.60 | 65.74 | 66.49 | 33.79 | 46.03 | 63.18 | 17.18 | 53.20 | 60.57 | 33.95 | 41.57 | 53.66 |
| | ✓ | 60.62 | 53.85 | 62.22 | 53.01 | 52.90 | 14.96 | 72.56 | 58.56 | 55.47 | 69.21 | 32.27 | 42.97 | 69.90 | 12.36 | 43.51 | 72.51 | 36.65 | 32.61 | 53.34 |
| PSAC | × | 53.01 | 56.68 | 57.41 | 48.12 | 49.55 | 26.43 | 72.96 | 60.69 | 50.56 | 55.54 | 41.98 | 52.30 | 56.70 | 19.58 | 38.13 | 58.92 | 26.68 | 46.24 | 50.45 |
| | ✓ | 55.14 | 52.26 | 64.70 | 44.45 | 52.34 | 22.15 | 72.06 | 60.70 | 58.97 | 52.35 | 41.18 | 49.78 | 53.28 | 18.85 | 42.60 | 64.53 | 25.81 | 45.68 | **51.31** |
| AVSD | × | 54.00 | 47.84 | 60.61 | 47.79 | 60.34 | 10.07 | 74.78 | 61.43 | 66.28 | 61.98 | 33.00 | 40.35 | 46.21 | 8.06 | 51.98 | 66.00 | 40.14 | 41.52 | 52.33 |
| | ✓ | 55.87 | 46.18 | 64.41 | 48.05 | 63.32 | 7.41 | 73.78 | 58.20 | 74.74 | 70.80 | 37.85 | 34.55 | 35.53 | 6.11 | 49.96 | 67.88 | 44.03 | 43.89 | **53.09** |
| LAViT | × | 50.57 | 43.45 | 50.78 | 44.93 | 47.28 | 15.50 | 67.19 | 65.51 | 52.37 | 22.04 | 44.35 | 61.69 | 52.21 | 21.52 | 45.61 | 40.49 | 35.00 | 49.33 | 47.40 |
| | ✓ | 45.05 | 45.09 | 57.33 | 41.26 | 48.62 | 17.00 | 69.91 | 65.90 | 60.61 | 29.57 | 43.17 | 57.57 | 53.92 | 22.09 | 54.46 | 35.35 | 33.99 | 49.40 | **48.91** |
| STG | × | 56.40 | 41.48 | 62.28 | 57.59 | 59.86 | 12.94 | 64.31 | 54.00 | 73.35 | 77.26 | 35.35 | 40.49 | 48.31 | 8.41 | 53.30 | 62.44 | 40.25 | 38.15 | 52.80 |
| LAVisH | × | 61.73 | 43.99 | 65.06 | 60.38 | 65.53 | 11.13 | 70.21 | 64.73 | 77.83 | 79.46 | 41.76 | 41.20 | 49.88 | 14.87 | 59.26 | 65.10 | 41.84 | 46.26 | 57.63 |
| | ✓ | 74.02 | 65.17 | 64.73 | 53.15 | 71.96 | 40.56 | 68.49 | 66.00 | 63.17 | 66.68 | 30.11 | 43.80 | 63.77 | 26.51 | 56.31 | 63.46 | 50.79 | 42.85 | **59.25** |
| Ours | ✓ | **84.32** | **67.23** | 64.68 | 62.18 | **75.09** | **48.42** | **80.47** | **66.38** | 77.22 | 67.58 | 55.15 | **82.23** | **70.12** | **39.83** | 61.26 | 58.17 | 43.67 | **58.33** | **66.95** |

## 5.4 Comparison on MUSIC-AVQA

We conduct experiments on the MUSIC-AVQA test split to validate the effectiveness of the proposed architecture. The results are presented in Table 1. *All methods, except LAVisH, utilize ResNet-18 to encode visual features, whereas LAVisH employs stronger models such as ViT [64] or Swin [65] to acquire visual representations.* We first analyze the comparison under the same visual encoding conditions. Notably, compared with COCA, our architecture obtains significant improvements of 4.53% and 4% in the audio and visual QA tasks, respectively. It also achieves the best performance in the AVQA task. Furthermore, our architecture obtains a new state-of-the-art result of 72.20% on the whole question. It is worth mentioning that visual QA methods, like GRU, and video QA methods, such as HME, also exhibit competitive results in the AVQA task, despite lacking one modality as input. We also observe that LAVisH, proposed based on STG, introduces trainable parameters into robust visual encoders, thereby achieving superior results compared to methods employing weaker visual encoders. The results on this dataset can demonstrate the efficacy of these methods to some extent. However, MUSIC-AVQA lacks refined and precise evaluations due to its inherent shortcomings that are introduced in Section 1. *Consequently, it is insufficient to evaluate these methods only on this dataset.*

## 5.5 Robustness Evaluation

We conduct experiments on the MUSIC-AVQA-R test split to explore the robustness of the aforementioned methods. Their released codes are employed to conduct this experiment. The results are presented in Table 2. Several crucial insights arise when combining the results from this table with those from Table 1. Firstly, audio QA methods, such as CONVLSTM, showcase competitive robustness and even achieve the highest tail accuracy on EXIST questions within the AVQA task. Secondly, the visual QA method MCAN demonstrates noteworthy robustness by obtaining the second-best performance on the test split. Thirdly, the video QA baseline experiences a relatively significant performance degradation, with PSAC, for instance, declining by 16.45%. Notably, HCRN exhibits the lowest performance on both datasets, indicating its poor robustness. *Furthermore, the results of AVQA baselines lag behind other types of QA methods like HME, suggesting that their strong performance on the MUSIC-AVQA dataset may rely on memorizing statistical regularities between input modalities and answers.* Ultimately, our architecture outperforms others on the test split. Benefiting from the MCCD strategy, it attains the highest head (in-distribution setting) and tail (out-of-distribution setting) results across various question types, providing further evidence of its superior robustness. We also show the overall accuracy of these methods on each type of question. Please see the details in Appendix D.4. It can be seen that our architecture achieves the best overall accuracy on each type of question.

To validate the plug-and-play capability of MCCD, we conduct extensive experiments using the *baselines+MCCD* on the aforementioned datasets. The results are presented in Tables 1 and 2. We observe that MCCD consistently improves performance across most methods on MUSIC-AVQA (9 out of 13) and MUSIC-AVQA-R (11 out of 13), respectively. This underscores the robust debiasing capability of MCCD in a plug-and-play manner.

## 5.6 Ablation Study

To verify the debiasing effectiveness of MCCD, we conduct extensive experiments on both the test split of MUSIC-AVQA and MUSIC-AVQA-R. The results are shown in Table 3. Firstly, we validate the contribution of the component within multi-faceted debiasing. It can be seen that removing the component will lead to an overall performance improvement in some aspects of MUSIC-AVQA while resulting in a significant decrease in our dataset. This observation

Table 3: Ablation results (%) on the test split of MUSIC-AVQA and our dataset. AQA and VQA denote audio QA, and visual QA, respectively. $d_i^{(\#)}$ is the distance between the (#) logit and the multi-modality logit. MD: multifaceted debiasing. CG: cycle guidance.

| Method | MUSIC-AVQA | | | | MUSIC-AVQA-R | | | | | |
|--------|------|-------|------|-------|------|-------|------|-------|-------|-------|
| | AQA | VQA | AVQA | All | AQA | VQA | AVQA | H | T | All |
| Ours | 79.14 | 78.24 | 67.13 | 72.20 | 74.76 | **72.76** | **61.34** | **72.24** | **59.39** | **66.95** |
| w/o $d_i^q$ | **79.64** | 77.62 | 67.52 | 72.34 | 74.89 | 72.26 | 59.53 | 71.72 | 57.47 | 65.86 |
| w/o $d_i^v$ | 79.27 | 78.61 | 67.23 | 72.37 | 71.29 | 70.18 | 58.08 | 68.34 | 57.43 | 63.85 |
| w/o $d_i^a$ | 77.22 | 77.50 | 67.31 | 71.76 | 70.92 | 67.82 | 55.57 | 66.05 | 55.61 | 61.75 |
| w/o MD | 78.46 | 77.54 | 66.39 | 71.48 | 73.97 | 70.41 | 58.87 | 70.09 | 57.25 | 64.80 |
| w/o CG | 78.77 | **78.65** | **67.50** | **72.45** | **75.42** | 71.72 | 59.68 | 71.40 | 57.98 | 65.87 |

strongly supports the debiasing efficacy of these components. Secondly, we verify the overall contribution of the multifaceted debiasing. It can be seen that the performance decrease of 0.72% and 2.15% occurs in both datasets, respectively. Finally, we validate the contribution of cycle guidance. We see that this model variant obtains the best performance on the MUSIC-AVQA dataset. However, there was a noticeable performance degradation in our proposed dataset. In summary, each component plays a distinctive role in the debiasing process, which is further demonstrated by the performance degradation on the head and tail samples.

## 5.7 Sensitivity and Qualitative Analysis

We employ the control variable method to perform a sensitivity analysis on the weight-controlling factors of the MCCD strategy. The results are presented in the left part of Fig. 5. Our findings indicate stable optimization across various settings, except for the case with $\alpha = 0.008$ and $\beta = 0.3$. Upon conducting further experimental analysis, we identify the issue as originating from the model's failure to converge. Moreover, we visualize the attention weight on the uniformly sampled audio and video frames to qualitatively analyze the debiasing capability of our method. The visualization, displayed in the right part of Fig. 5, reveals that crucial audio and video frames for QA consistently receive significant attention, both in in- and out-of-distribution settings. This further demonstrates that our method predicts answers through the grounding capabilities of audio and vision rather than relying on bias learning. More cases are shown in Appendix D.5.

## 6 Conclusion and Limitation

We are the first to investigate bias learning in the AVQA task from model evaluation and design aspects. On the one hand, we construct a new dataset, MUSIC-AVQA-R, which evaluates the performance on the head, tail, and overall samples, providing a precise measure of model robustness. On the other hand, we introduce a robust architecture employing the MCCD strategy to mitigate bias learning. Extensive experiments demonstrate the effectiveness of our architecture and the plug-and-play debiasing capability of MCCD. Furthermore, we reevaluate previous multi-modal QA methods on our proposed dataset, revealing their poor robustness.

Due to constraints imposed by MUSIC-AVQA, the answer space of our dataset is limited, comprising only 42 classes, and answer lengths are typically confined to a single word. This deviation from real-world scenarios is noteworthy. Concerning model designs, for a fair comparison with baselines, we do not select large generative models to be backbones. However, compared with the answer classification, it may be more useful to generate answers for the AVQA task.

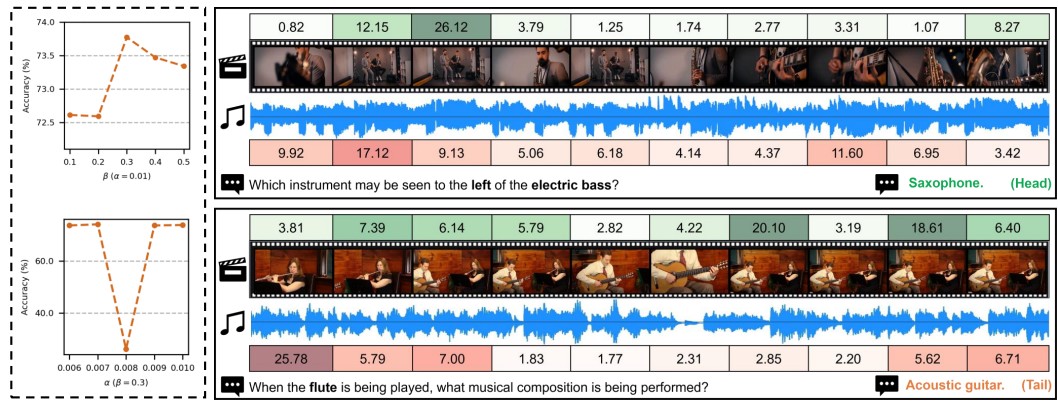

Figure 5: Sensitivity and qualitative analysis. $\alpha$ and $\beta$ are the weight-controlling factors in the MCCD strategy. We visualize attention weights on the uniformly sampled audio and video frames.

# 7 Acknowledgements

This work was supported by the National Key Research and Development Program of China (2021YFB1715600), the National Natural Science Foundation of China (U22B2019, 62477037, 62450005, 62437002, 62306229, 62293553), the Natural Science Basic Research Program of Shaanxi (2023-JC-YB-593), the Youth Innovation Team of Shaanxi Universities "Multi-modal Data Mining and Fusion", the Shaanxi Undergraduate and Higher Education Teaching Reform Research Program (23BY195), the Youth Talent Support Program of Shaanxi Science and Technology Association (20240113), the Xi'an Jiaotong University-China Mobile Communications Group Co., Ltd. Digital Government Joint Institute, and the China Postdoctoral Science Foundation (2024M752585).

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

# A  Model Training and Testing

The details of model training are shown in Algorithm 1, where $L$ denotes the number of training samples, and $n_{\mathrm{b}}$ is the batch size. In the test stage, the bias learner is removed.

---

**Algorithm 1:** Model Training

---

**Input:** $\mathcal{D} = \{(A_i, V_i, Q_i, y_i)\}_{i=1}^{L}$.
**Output:** Robust AVQA model.
1 Initialize model parameters $\theta$;
2 Initialize Adam optimizer;
3 Set learning rate $\eta$;
4 Set the number of training epochs $M$;
5 **for** *epoch* $\leftarrow 1$ *to* $M$ **do**
6     **for** *each batch in* $\mathcal{D}$ **do**
7        Learn uni-modal and multi-modal representations: $\mathbf{A}^{\mathrm{e}} \leftarrow \mathrm{VGGish}(A)$,
        $\mathbf{V}^{\mathrm{e}} \leftarrow \mathrm{ResNet18}(V)$, $\mathbf{A}^{\mathrm{c}} \leftarrow \mathrm{VisualBert}(\mathbf{A}^{\mathrm{e}})$, $\mathbf{V}^{\mathrm{c}} \leftarrow \mathrm{VisualBert}(\mathbf{V}^{\mathrm{e}})$,
        $\mathbf{Q}^{\mathrm{c}} \leftarrow \mathrm{VisualBert}(Q)$, $\mathbf{M}^{\mathrm{c}} \leftarrow \mathrm{VisualBert}([\mathbf{A}^{\mathrm{c}}|\mathbf{V}^{\mathrm{c}}], \mathbf{Q}^{\mathrm{c}})$;
8        Capture uni-modal biases: $\hat{\mathbf{y}}^{\mathrm{a}} \leftarrow \mathrm{BiasLearner}_{\mathrm{a}}(\mathbf{A}^{\mathrm{c}})$, $\hat{\mathbf{y}}^{\mathrm{v}} \leftarrow \mathrm{BiasLearner}_{\mathrm{v}}(\mathbf{V}^{\mathrm{c}})$,
        $\hat{\mathbf{y}}^{\mathrm{q}} \leftarrow \mathrm{BiasLearner}_{\mathrm{q}}(\mathbf{Q}^{\mathrm{c}})$;
9        Obtain answer predictions: $\hat{\mathbf{y}}^{\mathrm{m}} \leftarrow \mathrm{Classifier}(\mathbf{M}^{\mathrm{c}})$;
10       Compute the QA loss: $\mathcal{L}_{\mathrm{a}} \leftarrow -\frac{1}{n_{\mathrm{b}}} \sum \mathbf{y} \log \hat{\mathbf{y}}^{\mathrm{m}}$;
11       Mitigate biases by MCCD: $\mathcal{L}_{\mathrm{d}}, \mathcal{L}_{\mathrm{c}} \leftarrow \mathrm{MCCD}(\hat{\mathbf{y}}^{\mathrm{a}}, \hat{\mathbf{y}}^{\mathrm{v}}, \hat{\mathbf{y}}^{\mathrm{q}}, \hat{\mathbf{y}}^{\mathrm{m}})$;
12       Compute the joint loss: $\mathcal{L} \leftarrow \mathcal{L}_{\mathrm{a}} + \mathcal{L}_{\mathrm{d}} + \mathcal{L}_{\mathrm{c}}$;
13       Backward pass: $\nabla\theta \leftarrow \nabla_{\hat{\mathbf{y}}^{\mathrm{a}}, \hat{\mathbf{y}}^{\mathrm{v}}, \hat{\mathbf{y}}^{\mathrm{q}}, \hat{\mathbf{y}}^{\mathrm{m}}} \mathcal{L}$;
14       Update model parameters: $\theta \leftarrow \mathrm{Optimize}(\theta, \nabla\theta, \eta)$;
15 **return** *AVQA model.*

---

# B  Dataset Analysis

We make statistics on the rephrasings, as depicted in Table 4. Any rephrasing that garners fewer than one vote will be disregarded. It is evident that the overwhelming majority of the rephrased questions received three favorable votes. Fig. 6 shows the distribution of questions in MUSIC-AVQA and

Table 4: Statistics of rephrasing consistency. Positive and Negative denote whether the annotator agrees with the rephrasing or not.

| Positive | Negative | Total |
|:---:|:---:|:---:|
| 3 | 0 | 164, 219 |
| 2 | 1 | 47,353 |
| 1 | 2 | 7,481 |
| 0 | 3 | 9,172 |

MUSIC-AVQA-R based on the first three words. It is evident that there are notably more entries within each circle in the left figure compared to the right figure. This suggests that the diversity of our dataset is higher than MUSIC-AVA.

We visualize the answer distribution of specific types of questions. Fig. 7, 8, and 9 show the visualization of the AVQA, audio QA and visual QA tasks, respectively. We can see that all of the answer distributions are long-tail. This further demonstrates the necessity of head and tail sample splitting. Specifically, for the "Location" type in the AVQA task, we can see that the number of "congas" is far less than that of "yes". It is noteworthy that, for question types featuring only two possible answers, we classify questions with lower frequencies as tail samples and those with higher frequencies as head samples. For instance, for the "Existential" questions in the AVQA task, we consider the questions with the answer "no" as tail samples.

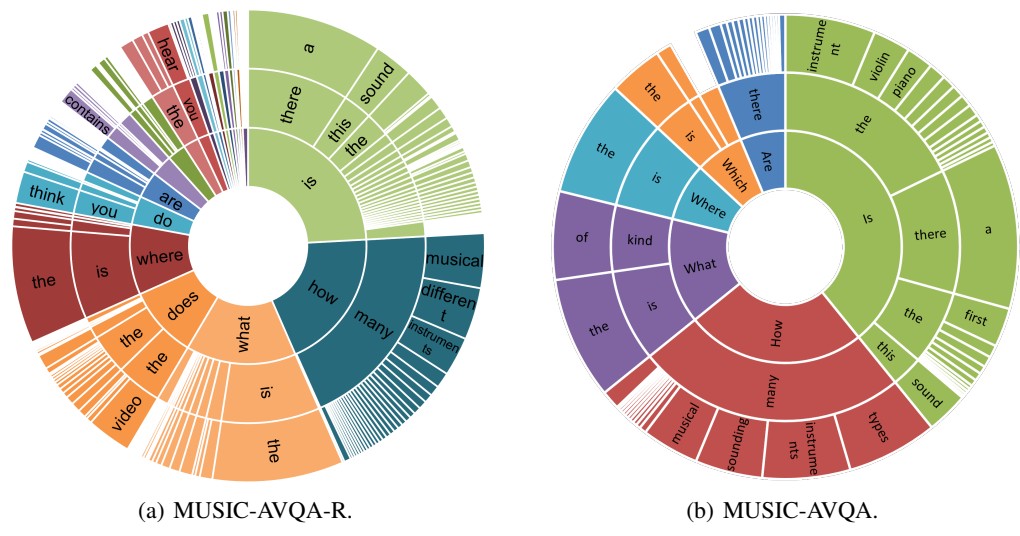

(a) MUSIC-AVQA-R.          (b) MUSIC-AVQA.

Figure 6: Distribution visualization of questions based on the first three words.

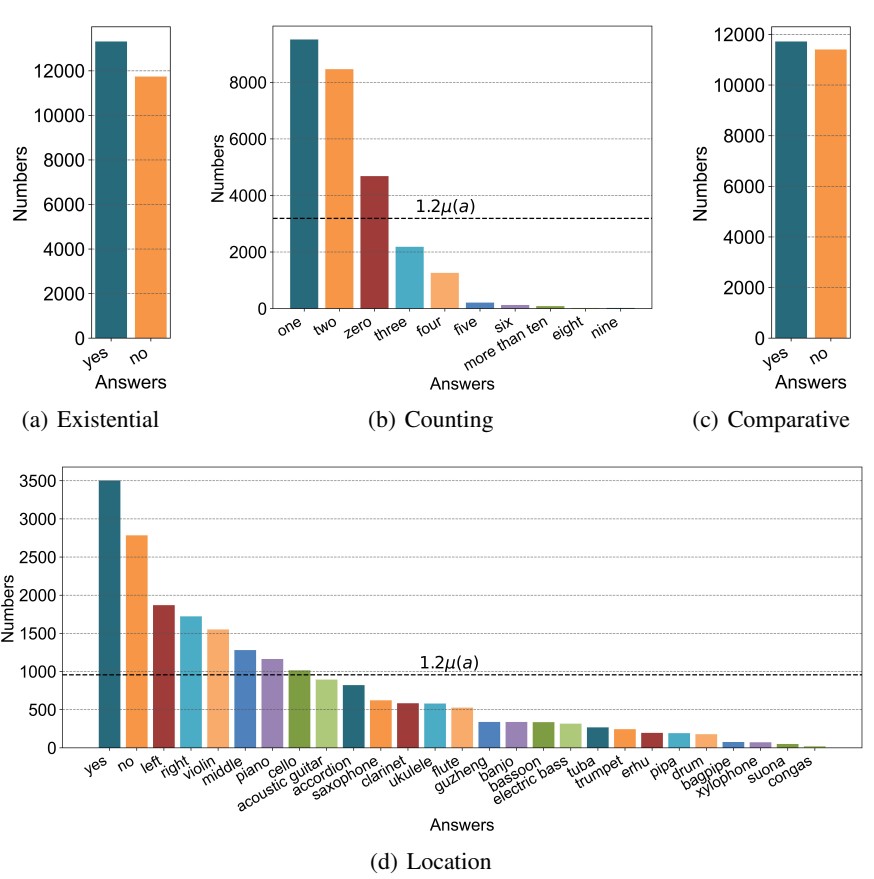

(a) Existential          (b) Counting          (c) Comparative

(d) Location

Figure 7: Answer distributions of specific types of questions in the AVQA task. $\mu(a)$ is the average number of answers in a group.

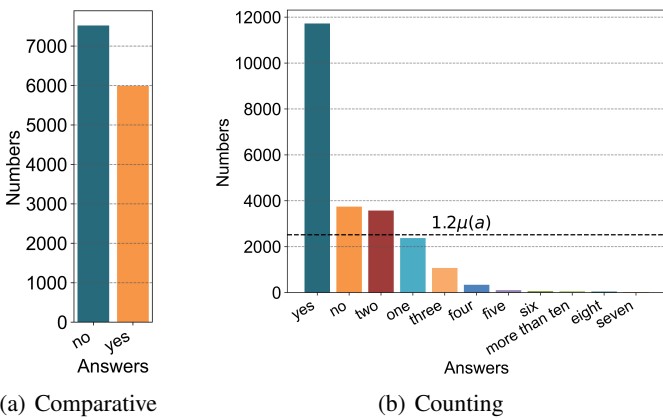

(a) Comparative

(b) Counting

Figure 8: Answer distributions of specific types of questions in the audio QA task. $\mu(a)$ is the average number of answers in a group.

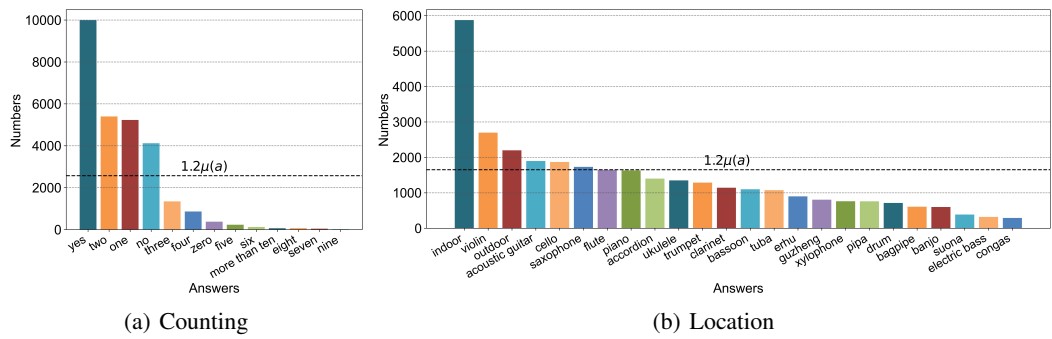

(a) Counting

(b) Location

Figure 9: Answer distributions of specific types of questions in the visual QA task. $\mu(a)$ is the average number of answers in a group.

## B.1 Ethics Statement

MUSIC-AVQA [4] has undergone meticulous pre-processing tailored for academic research purposes. We ensure the absence of any information that discloses the names or uniquely identifies individuals, as well as avoiding any offensive content. We only perform rephrasing and splitting for the questions within the dataset to develop MUSIC-AVQA-R, which preserves its inherent characteristics.

## B.2 Question Comparison

We select questions from both the head and tail splits to demonstrate the diversity of our dataset, as illustrated in Figures 10 and 11. Due to the use of pre-defined templates, the questions in the train and test splits of MUSIC-AVQA differ by only a single word. In contrast, the questions in our dataset exhibit a variety of formats, which better reflect real-world scenarios. Additionally, our dataset encompasses a larger test space compared to MUSIC-AVQA.

> **Predefined Question Template in MUSIC-AVQA:**
> ◆ Which is the musical instrument that sounds at the same time as the <Object>?
>
> **Question in MUSIC-AVQA:**
> ☐ Which is the musical instrument that sounds at the same time as the pipa?
>
> **Rephrased Question in MUSIC-AVQA-R (Head Split):**
> ✓ Which is the musical instrument that sounds at the same time as the pipa?
> ✓ Which musical instrument plays simultaneously with the pipa?
> ✓ What musical instrument is playing in unison with the pipa?
> ✓ What musical piece is playing in the background as the pipa is being played?
> ✓ What musical instrument is playing concurrently with the pipa?
> ✓ What musical instrument is being played when the pipa is being played?
> ✓ What musical piece is being played simultaneously with the pipa?
> ✓ When the pipa is being played, what musical composition is being performed?
> ✓ What musical work is being performed when the pipa is being played?
> ✓ The pipa is being played, but what musical composition is that?
> ✓ When playing the pipa what musical composition is being performed?
> ✓ What instrument is playing simultaneously with the pipa?
> ✓ What other instrument is playing at the same time as the pipa?
> ✓ What musical instrument plays simultaneously with the pipa?
> ✓ What is the musical instrument that produces sound in tandem with the pipa?
> ✓ What musical instrument produces sound simultaneously with the pipa?
> ✓ ......

Figure 10: Question Comparison between MUSIC-AVQA and MUSIC-AVQA-R. The rephrased question comes from the head split of our dataset. The questions in our dataset feature diverse formats, which more accurately reflect real-world scenarios.

## C Proof

Given a probability density function $f(x)$ that conforms to a uniform distribution with size $N$, its entropy $H(X)$ can be calculated as follows:

$$H(X) = -\sum_{i=1}^{N} p(x_i) \cdot \log_2 p(x_i), \tag{3}$$

$$p(x_i) = f(x_i), \tag{4}$$

where $p(x_i)$ is the probability of $X = x_i$.

In a uniform distribution, each probability $p(x_i)$ is the same, *i.e.*, $p(x_i) = f(x_i) = \frac{1}{N}$, so

$$H(X) = -\sum_{i=1}^{N} \frac{1}{N} \cdot \log_2 \left( \frac{1}{N} \right), \tag{5}$$

Then, bring the $\frac{1}{N}$ term outside the summation:

$$H(X) = -\frac{1}{N} \sum_{i=1}^{N} \log_2 \left( \frac{1}{N} \right), \tag{6}$$

Thirdly, move the negative sign inside the logarithm:

$$H(X) = \frac{1}{N} \sum_{i=1}^{N} \log_2(N), \tag{7}$$

Finally, combine $\frac{1}{N}$ with the summation:

$$H(X) = \log_2(N). \tag{8}$$

Figure 11: Question Comparison between MUSIC-AVQA and MUSIC-AVQA-R. The rephrased question comes from the tail split of our dataset. The questions in our dataset feature diverse formats, which more accurately reflect real-world scenarios.

# D Experiments

## D.1 Dataset Comparison

The test split comparison between MUSIC-AVQA and MUSIC-AVQA-R is shown in Table 5. We can see that our proposed dataset exhibits a larger test sample space. This can provide a more precise evaluation for the model robustness.

Table 5: Test split comparison between MUSIC-AVQA and MUSIC-AVQA-R. EXIST, LOC, CNT, COMP, and TEMP, which are question types, denote "Existential", "Location", "Counting", "Comparative", and "Temporal", respectively.

| Dataset | Audio QA | | Visual QA | | AVQA | | | | |
|---|---|---|---|---|---|---|---|---|---|
| | CNT | COMP | CNT | LOC | EXIST | LOC | CNT | COMP | TEMP |
| MUSIC-AVQA | 1,017 | 594 | 1,197 | 1,225 | 988 | 920 | 1,265 | 1,101 | 822 |
| MUSIC-AVQA-R | 23,107 | 13,506 | 27,867 | 3,3049 | 25,049 | 21,546 | 26,565 | 23,121 | 17,762 |

## D.2 Implementation Details

The number of trainable parameters of our model is 117M. In the default settings, our model training takes about 20 hours. We initialize the seed for both Numpy and Torch to 42. The other details can be found in our uploaded code.

## D.3 Baselines

The audio QA baselines are as follows.

- **FCNLSTM**[2] employs a fully convolutional network and LSTM to initially learn the representations of audio and questions separately. Subsequently, it projects the concatenated features of both into the answer space.
- **CONVLSTM** is a variant of FCNLSTM, incorporating five convolutional blocks identical to VGGNet for acquiring a variable-sized representation of audio.

The visual QA baselines are as follows.

- **GRU** (dubbed "deeper LSTM + Norm I" in the published paper) is a simple baseline that first uses VGGNet and LSTM to encode the images and questions and then maps the concatenated features of them into the answer space.
- **BiLSTM Attn** is an attention-based bi-directional LSTM network, which was often used in previous relation classification.
- **HCAttn**[3] is a hierarchical co-attention method that employs question- and image-guided attention to reason on images and questions, respectively.
- **MCAN**[4] is a deep modular co-attention network comprised of cascaded modular co-attention layers, where attention is implemented through multi-head attention in Transformers.

The video QA baselines are as follows.

- **HME**[5] is a heterogeneous memory-enhanced multimodal attention model that can effectively learn global context information from appearance and motion features.
- **PSAC**[6] employs positional self-attention block to model the dependency between question words and video frames, respectively. It utilizes a co-attention mechanism to perform multi-modal interaction.
- **HCRN**[7] is a hierarchical conditional relation network that embeds video input at various granularities, encompassing frames, short clips, and entire video levels.

The AVQA baselines are as follows.

- **AVSD**[8] is a simple but effective audio-visual dialog method. It initially encodes the input modalities separately and subsequently feeds their fused features into a LSTM to generate answers.
- **LAViT**[9] is a spatial AVQA framework that utilizes three distinct Transformer blocks to perform interaction between input modalities.
- **STG**[10] associates particular visual locations with audio to conduct spatial grounding. Based on this, audio and visual features of key timestamps are further emphasized through question queries for temporal grounding.
- **LAVisH**[11], based on STG, incorporates trainable parameters into powerful visual encoders such as ViT and Swin.

### D.4 Reevaluation on MUSIC-AVQA-R

Table 6 illustrates the overall accuracy for specific types of questions. Notably, our architecture surpasses all baselines across every question type. Importantly, our architecture achieves the highest performance in all three tasks, underscoring the idea that the additional modality serves as a valuable complement. For instance, the audio modality may enhance AVQA models in the video QA task.

---

[2] https://github.com/facebookresearch/daqa
[3] https://github.com/jiasenlu/HieCoAttenVQA
[4] https://github.com/MILVLG/mcan-vqa
[5] https://github.com/fanchenyou/HME-VideoQA
[6] https://github.com/lixiangpengcs/PSAC
[7] https://github.com/thaolmk54/hcrn-videoqa
[8] https://github.com/idansc/simple-avsd
[9] https://github.com/hs-yn/PanoAVQA
[10] https://github.com/GeWu-Lab/MUSIC-AVQA
[11] https://github.com/GenjiB/LAVISH

Table 6: Experimental results on the MUSIC-AVQA-R test split. The numerical values represent the overall accuracy for specific types of questions. EXIST, LOC, CNT, COMP, and TEMP are question types, representing "Existential", "Location", "Counting", "Comparative", and "Temporal", respectively.

| Type | Method | Audio QA | | | Visual QA | | | AVQA | | | | | | All |
| --- | --- | --- | --- | --- | --- | --- | --- | --- | --- | --- | --- | --- | --- | --- |
| | | CNT | COMP | Avg. | CNT | LOC | Avg. | EXIST | LOC | CNT | COMP | TEMP | Avg. | Avg. |
| Audio QA | FCNLSTM | 60.98 | 58.74 | 60.15 | 55.41 | 52.93 | 54.07 | 71.21 | 48.67 | 44.76 | 57.21 | 34.34 | 52.21 | 54.12 |
| | CONVLSTM | 65.09 | 61.04 | 63.60 | 56.17 | 51.63 | 53.71 | 71.48 | 48.36 | 43.18 | 57.71 | 43.08 | 53.31 | 55.20 |
| Visual QA | BiLSTM Attn | 68.85 | 46.36 | 60.56 | 57.08 | 47.87 | 52.08 | 56.41 | 36.79 | 43.23 | 50.68 | 23.49 | 43.34 | 48.84 |
| | HCAttn | 58.13 | 53.79 | 56.53 | 51.21 | 59.98 | 55.97 | 64.02 | 38.08 | 44.32 | 54.38 | 36.16 | 48.24 | 51.90 |
| | GRU | 63.69 | 58.88 | 61.92 | 58.46 | 57.67 | 58.03 | 70.89 | 41.91 | 47.12 | 57.29 | 35.16 | 51.56 | 55.21 |
| | MCAN | 72.40 | 54.99 | 65.98 | 60.33 | 64.33 | 62.50 | 58.78 | 49.04 | 51.99 | 51.47 | 44.71 | 51.69 | 57.27 |
| Video QA | HCRN | 55.38 | 41.28 | 50.18 | 39.89 | 45.33 | 42.84 | 53.07 | 38.00 | 42.80 | 35.03 | 42.32 | 37.94 | 43.92 |
| | PSAC | 53.66 | 53.29 | 53.52 | 46.95 | 66.73 | 57.68 | 52.89 | 43.94 | 44.70 | 48.38 | 35.09 | 45.61 | 50.45 |
| | HME | 61.07 | 56.44 | 59.36 | 47.08 | 65.81 | 57.24 | 66.09 | 36.11 | 48.31 | 56.84 | 37.23 | 49.91 | 53.66 |
| AVQA | LAViT | 49.07 | 48.19 | 48.74 | 43.71 | 66.33 | 55.98 | 38.16 | 47.63 | 42.29 | 43.09 | 41.16 | 42.38 | 47.40 |
| | AVSD | 52.91 | 54.92 | 53.66 | 54.70 | 68.01 | 61.92 | 64.26 | 34.39 | 33.88 | 58.89 | 40.37 | 46.79 | 52.33 |
| | STG | 53.77 | 60.20 | 56.14 | 54.59 | 59.07 | 57.02 | 75.18 | 36.33 | 35.41 | 57.81 | 39.35 | 49.47 | 52.80 |
| | **Ours** | **81.30** | **63.57** | **74.76** | **72.09** | **73.32** | **72.76** | **75.36** | **57.37** | **60.32** | **59.74** | **49.97** | **61.34** | **66.95** |

## D.5 Case Study

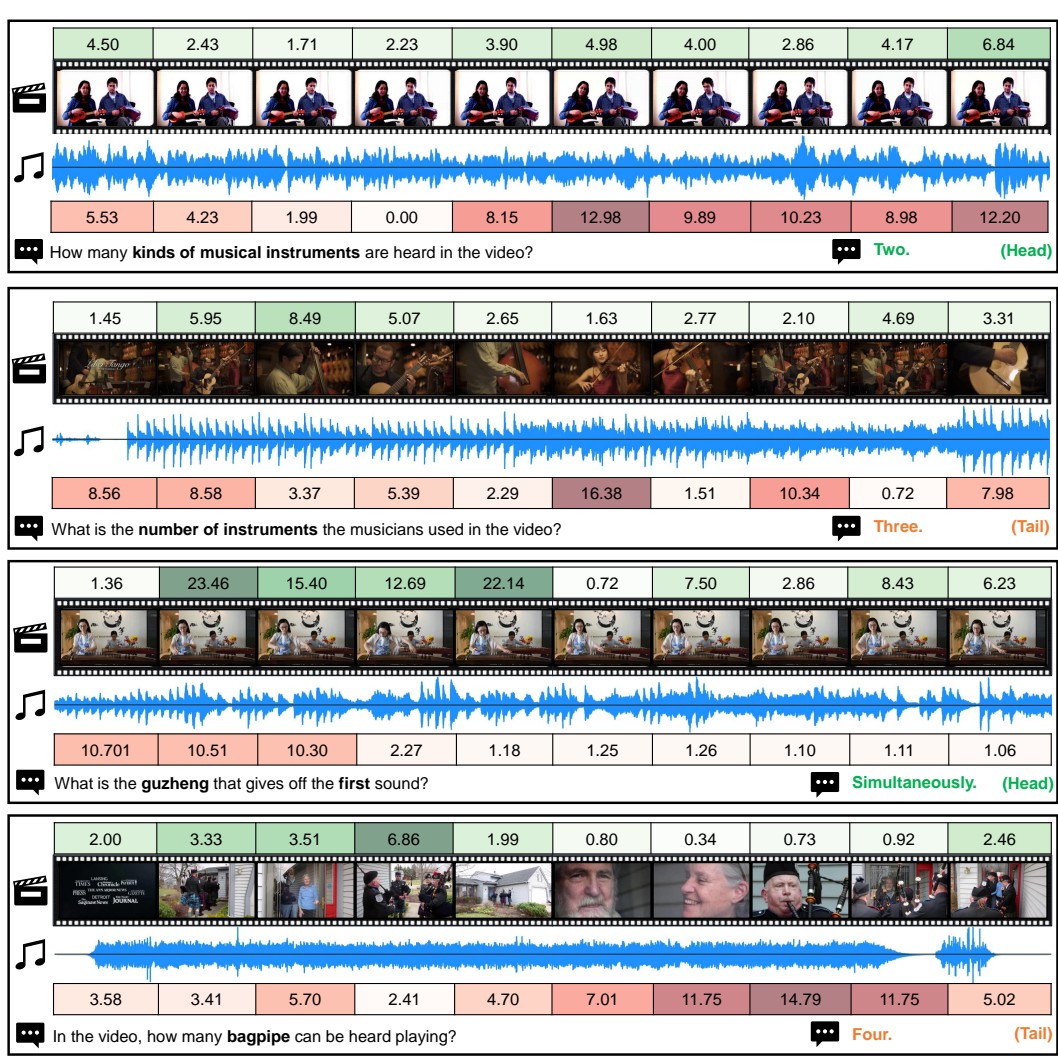

Figure 12: Attention weight visualization on the uniformly sampled audio and video frames.

We visualize attention weight on extra uniformly sampled audio and video frames to qualitatively analyze the debiasing capability. In Fig. 12, the visualization is presented for a more extensive range of head and tail samples. We can see that our method can focus on the key audio and video frames for QA simultaneously in both in- and out-of-distribution settings. For instance, in the upper head and tail case, our method demonstrates high attention to various instruments, leading to accurate answers for "counting" questions. This serves as additional evidence supporting the debiasing effectiveness of our proposed MCCD strategy, highlighting its substantial contribution to improving model robustness.

