# OpenReview forum: "Look, Listen, and Answer: Overcoming Biases for Audio-Visual Question Answering"
_NeurIPS.cc/2024/Conference — NeurIPS 2024 poster_

### Official Review · Reviewer_npgc · 2024-06-18

**Soundness:** 4
**Presentation:** 4
**Contribution:** 3
**Rating:** 8
**Confidence:** 5

**Summary:**

The authors have made a series of contributions to the field of Audio-Visual Question Answering (AVQA). Firstly, the introduction of the innovative dataset MUSIC-AVQA-R, along with its associated evaluation metrics, stands out. This dataset and metrics are crucial for a comprehensive evaluation of AVQA models' reasoning capabilities and their ability to generalize across different scenarios. Secondly, their novel AVQA architecture, which incorporates the MCCD strategy to mitigate training biases, is interesting. This marks the first concerted effort to systematically tackle biases in AVQA tasks, examining both model evaluations and designs. Additionally, the authors have undertaken extensive experiments with both MUSIC-AVQA and MUSIC-AVQA-R, providing robust evidence of the proposed architecture and debiasing strategy's effectiveness and superiority. Finally, their evaluation of 13 multimodal QA methods on the new dataset exposes a significant limitation: these methods struggle to generalize effectively, whether within the distribution or in out-of-distribution contexts.

**Strengths:**

(1) The robustness of multimodal models is worth exploring in depth. The authors are inspired by robust VQA and propose the first AVQA dataset and corresponding evaluation metrics for robust AVQA. This dataset has a large number of test samples and can provide fine-grained evaluations for both head and tail samples, which is very interesting. The authors also conduct a detailed analysis for this dataset. I think the quality of this dataset would be acknowledged by the community.

(2) The authors suppose that completely avoiding the negative bias in the dataset is challenging. The development of this dataset encourages the community to enhance the model's robustness through the implementation of debiasing strategies. To this end, the authors proposed an AVQA architecture that incorporates the MCCD strategy to overcome the bias. The authors not only explore biases in the AVQA task from the perspective of model evaluations but also from model designs.

(3) The authors carried out extensive experiments on MUSIC-AVQA and MUSIC-AVQA-R to demonstrate the effectiveness and superiority of their proposed architecture and the MCCD strategy. The experimental results in Table 1 and 2 demonstrate the effectiveness of the architecture. In addition, the results also demonstrate the plug-and-play capability of the proposed MCCD strategy. The results in Table 3 demonstrate the effectiveness of the component within the MCCD strategy. I think the experimental results can strongly support their claim for the architecture and MCCD strategy.

(4) The authors evaluate 13 recent multimodal QA methods on the proposed dataset and show their limited ability to generalize not only in-distribution scenarios but also in out-of-distribution situations.  These experiments further strengthen the solidity of this work.

(5) The authors provide detailed descriptions of the dataset and the implementation settings of the architecture. Notably, the authors provide a detailed readme.md file for their uploaded dataset and code.

**Weaknesses:**

(1) In the dataset developing stage, the authors perform rephrasing for the original question 25 times. For example, given a question in the Appendix B.2 "Which is the musical instrument that sounds at the same time as the pipa?, the authors may obtain 25 new questions in the MUSIC-AVQA-R. I think, in the test stage, the evaluation metrics would be better considering how much a model can make correct predictions for the mentioned group. For example, if a AVQA model can only provide 5 accurate predictions for the 25 questions, you should devise a more fine-grained evaluation metric to evaluate the robustness of AVQA models.

**Questions:**

(1) Please see the weaknesses.

(2) I notice that the answer of MUSIC-AVQA is only single word. I mean that the answer of MUSIC-AVQA-R is still a single word. However, I think this may not be in line with the real-world scenario. Because I think the machine may generate a few words rather than a single word.  Which perspectives do you think should be taken to solve it?

(3) There is no more fine-grained split of bias types in the dataset. How do you think we should deal with it?

**Limitations:**

N/A.

---

> ### Author Rebuttal · Authors · 2024-08-05
>
> **Reply for Q1.** For a truly robust AVQA model, it should accurately answer questions that have the same meaning but are presented in different forms. In future work, we will devise more fine-grained evaluation metrics. The metric is as follows:
> $$
> ACC=\frac{\sum_{i=1}^{n_g} \operatorname{sgn}(i) }{n_g}, \\\\
> \operatorname{sgn}(i) =
> \begin{cases}
> 1 & \text{if } c_i \geq \frac{n_{i}}{2}, \\\\
> 0 & \text{else } c_i < \frac{n_{i}}{2},
> \end{cases}
> $$
>
> where $n_g$ is the number of question groups, n_i denotes the number of rephrased questions in the $i$-th group, $c_i$ is the number of questions accurately answered by AVQA models. In this equation, we consider an AVQA model to be robust if it correctly answers at least half of the total questions in a group.
>
> **Reply for Q2.** We believe that there are two potential solutions to address this issue: dataset construction and model design. First, we can build a small instruction-tuning dataset to guide AVQA models in generating accurate answers. Second, we can choose a generative model, such as AnyGPT [1], as the backbone. This backbone can be trained on the instruction dataset, making it seamlessly applicable to the AVQA task. To mitigate training bias, we can input the uni-modal token representation into MCCD. We will explore the issue you kindly pointed in future work.
>
> **Reply for Q3.** We believe that the bias in datasets can be explicitly collected. For example, we can feed only the audio modality into an AVQA model to predict answers and collect samples that are answered accurately. These samples can then serve as a test bed for audio bias mitigation. This process can also be applied to obtain video and question bias data. By doing so, we can evaluate AVQA models in a more fine-grained manner. We will explore this approach in future work.
>
> [1] Zhan, Jun, et al. "Anygpt: Unified multimodal llm with discrete sequence modeling." arXiv preprint arXiv:2402.12226 (2024).

---

> > ### Comment · Reviewer_npgc · 2024-08-09
> >
> > Thank you for your reply. Your response has effectively addressed my confusion and concerns. I also notice the explanations in the global reply. I believe the theoretical analysis is essential, as it enhances the depth of the manuscript. It would be beneficial to include this analysis in the appendix of the revised version. As a result, I have improved the ratings for this manuscript.

---

> ### Author Response · Authors · 2024-08-10
> **Thanks for your recognition.**
>
> First, we greatly appreciate your suggestions regarding fine-grained evaluation metrics, generative debiasing AVQA models, and more detailed bias collections. These ideas are highly valuable and will undoubtedly contribute to future work.
>
> Second, the inclusion of additional theoretical analyses for Equation (1) and the experimental results without training MCCD will significantly enhance the depth of our method, both theoretically and empirically. We will be incorporating them into the appendix of our manuscript.
>
> Thank you once again for your thorough and thoughtful review of our work.

---

### Official Review · Reviewer_b8Yd · 2024-07-13

**Soundness:** 1
**Presentation:** 2
**Contribution:** 2
**Rating:** 5
**Confidence:** 4

**Summary:**

Prevalent Audio-Visual Question Answering (AVQA) methods often learn biases from the dataset, which can lead to reduced robustness. Additionally, current datasets may not accurately assess these methods. To address these issues, the authors introduce MUSIC-AVQA-R, a dataset created in two steps. First, they rephrase questions in the test split of a public dataset (MUSIC-AVQA) and then introduce distribution shifts to the split questions. Rephrasing the questions allows for a diverse test space, while the distribution shifts enable a comprehensive evaluation of robustness for rare, frequent, and overall questions. The authors also propose a robust architecture that uses a multifaceted cycle collaborative debiasing strategy to overcome biased learning.

**Strengths:**

- The authors raised a new problem, the bias problem in audio-visual question answering.
- The authors proposed a new dataset.

**Weaknesses:**

- First, I have a concern about the quality of the new dataset, it is not verified enough whether the dataset is properly collected. Similar to the case of the VQA-CP dataset, the authors should claim that the proposed dataset is a useful testbed to measure the distribution shift out of distribution robustness by showing the distribution difference between the train and test datasets. However, the authors only show the distribution of the rephrased dataset without showing the distribution difference between the train and test datasets. In short, there is no way to see whether the collected data after paraphrasing and distribution shifting is a valid one or not. In other words, how does the resulting distribution differ from the previous distribution?
The proposed solution also lacks novelty. Extensive work has been done to mitigate modality bias in the VQA-CP dataset, such as [1,2,3]. Therefore, in order to study the modality bias problem in AVQA, the authors should also devise a novel solution specialized for the AVQA problem by comparing it with the existing debiasing methods for VQA.

[1] Counterfactual Samples Synthesizing for Robust Visual Question Answering. CVPR 2020.

[2] Greedy Gradient Ensemble for Robust Visual Question Answering. ICCV 2021.

[3] Generative Bias for Robust Visual Question Answering. CVPR 2023.

- In addition, the authors should discuss why the proposed method is better than existing solutions theoretically. However, there is no explanation for why simply increasing the logit distance among the modalities in Equation 1 reduces the bias. Is there any reference from which the authors adopted the ideas? What I especially don't understand is whether there is any theoretical evidence as to why it is necessary to add the inverse of distance. How about minus? More theoretical discussions are required.
- In addition, the proposed solution itself also doesn’t make sense. Equation 3, the KL divergence between logits, and equation 2, the Euclidean distance, are the same in that they measure the distance between the logits. In theory, when the temperature term in the softmax function is large enough, the kl divergence loss becomes equivalent to the Euclidean distance. It is unclear what is the meaning of putting these two similar losses in opposite directions? The authors should further explain the theoretical motivation for using those loss functions.
- The performance in Tables 1 and 2 is also questionable. If other existing debiasing methods, such as [1,2,3], were applied to the same baseline, could the proposed method still outperform them? The proposed solution will be meaningful only if the proposed method sufficiently outperforms the existing methods.
I don't understand how a simple method in this paper (ours in Table 1) achieves a favorable performance in MUSIC-AVQA with such a simple baseline architecture. What is the baseline model performance without adding the proposed loss functions? In Table 3, how does the model without any proposed loss function already show a favorable performance that outperforms the state-of-the-art methods?

**Questions:**

Please refer to the questions in the weakness part.

**Limitations:**

I can’t find a negative societal impact of this work.

---

> ### Author Rebuttal · Authors · 2024-08-05
>
> We appreciate your constructive comments and valuable suggestions.
>
> * **Reply for Q1**. Please see the above author's rebuttal attached with a "response.pdf".
> * **Reply for Q2**. Please see the above author's rebuttal attached with a "response.pdf".
> * **Reply for Q3**. We will explain the MCCD from theoretical and empirical views. **Theoretical explanation:** (1) different roles and purposes of the loss functions. **In Equation (1), we use the Euclidean distance to enlarge the dissimilarity between *uni-modal and multi-modal logits*.** Increasing these distances reduces the model's reliance on single-modal information, thereby mitigating bias. **In Equation (2), we use KL divergence to constrain the distribution of *uni-modal* logits.** By minimizing the KL divergence between uni-modal logits, we enforce a closer distribution, establishing a collaborative relationship among them. It may be obvious that relying solely on one modality for answer prediction may result in similar logit distribution because, for a multimodal task, we cannot choose a correct answer only given a single modality. Therefore, we employ KL divergence to constrain the distribution of uni-modal logits including $KL(\hat{\mathbf{y}} _ q, \hat{\mathbf{y}} _ a), KL(\hat{\mathbf{y}} _ a, \hat{\mathbf{y}} _ v), KL(\hat{\mathbf{y}} _ v, \hat{\mathbf{y}} _ q)$, where $\hat{\mathbf{y}} _ q$, $\hat{\mathbf{y}} _ a$, and $\hat{\mathbf{y}} _ v$ denote question, audio, and video bias logits respectively. These constraints drive the mentioned uni-modal logits to be similar. **In other words, Equation (2) only constrains the distribution of uni-modal logits, without involving the multimodal logits.** Therefore, Equation (1) and (2) are different and they do not cause opposite directions. **Experimental validation.** We demonstrate the effectiveness of the combined use of these two loss functions through ablation experiments. The results show a performance drop when either $\mathcal{L} _ {\mathrm{d}}$ or $\mathcal{L} _ {\mathrm{c}}$ is used alone, demonstrating the importance and necessity of their combined use. **In summary**, by combining $\mathcal{L} _ {\mathrm{d}}$ and $\mathcal{L} _ {\mathrm{c}}$, we can more effectively reduce bias and enhance collaboration between uni-modal logits, improving the model robustness. We hope the above explanation clarifies your concerns.
> * **Reply for Q4.** **First**, the goal of this manuscript is to provide a robust evaluation dataset and a bias elimination method for the AVQA task. Please see the four-fold contributions of this manuscript. We have clarified the differences between bias mitigation methods for VQA and AVQA tasks (see R1). **Second**, our model is based on the pre-trained model VisualBERT, which has been demonstrated higher robustness compared to smaller models, as shown in Table 7 of [3]. Specifically, LXMERT+GenB outperforms the VQA baseline SAN+GenB by 14.44%. The model performance without MCCD training on two datasets is detailed in our response to Reviewer Zk4k. **Additionally**, we have provided the dataset, code, and a detailed "readme" file. We believe these resources will address your concerns.
>
> [4] Anderson, Peter, et al. "Bottom-up and top-down attention for image captioning and visual question answering." Proceedings of the IEEE conference on computer vision and pattern recognition. 2018.
>
> [5] Cadene, Remi, et al. "Rubi: Reducing unimodal biases for visual question answering." Advances in neural information processing systems 32 (2019).
>
> [6] Li, Guangyao, et al. "Learning to answer questions in dynamic audio-visual scenarios." Proceedings of the IEEE/CVF Conference on Computer Vision and Pattern Recognition. 2022.
>
> [7] Lao, Mingrui, et al. "Coca: Collaborative causal regularization for audio-visual question answering." Proceedings of the AAAI Conference on Artificial Intelligence. Vol. 37. No. 11. 2023.
>
> [8] Ragonesi R, Volpi R, Cavazza J, et al. Learning unbiased representations via mutual information backpropagation. In: Proceedings of the IEEE/CVF conference on computer vision and pattern recognition. 2021: 2729-2738.
>
> [9] Zhu W, Zheng H, Liao H, et al. Learning bias-invariant representation by cross-sample mutual information minimization. In: Proceedings of the IEEE/CVF International Conference on Computer Vision. 2021: 15002-15012.

---

> ### Comment · Area_Chair_FGKi · 2024-08-12
> **Final rating**
>
> Hi Reviewer b8Yd,
>
> The discussion period is ending soon. Please take a moment to review the author's rebuttal and share any remaining concerns or questions.
>
> Thank you,
> AC

---

> ### Comment · Reviewer_b8Yd · 2024-08-12
> **Response to the authors**
>
> The authors have addressed my concerns. I raised the score accordingly.

---

> > ### Author Response · Authors · 2024-08-12
> > **Letter of Thanks**
> >
> > Dear reviewer,
> >
> > Thanks for your effort in reviewing our manuscript.
> >
> > Best Regards,

---

### Official Review · Reviewer_Zk4k · 2024-07-18

**Soundness:** 2
**Presentation:** 2
**Contribution:** 3
**Rating:** 6
**Confidence:** 3

**Summary:**

This paper first analyzes the bias in the existing benchmark (MUSIC-AVQA) on Audio-Visual Question Answering (AVQA). They find MUSIC-AVQA testing set is created using templates and only has 9k QA pairs with limited vocabulary, leading to the potential correlation between words in questions and answers. Therefore, previous methods optimized on MUSIC-AVQA could "memorize" the regularity between questions and answers and may not be proper for AVQA question in real world, which include diverse QA pairs. To evaluate the robustness of the AVQA methods, they use a rephraser to rephrase questions in MUSIC-AVQA testing set, thus create the MUSIC-AVQA-R test set. In addition to the rephrased testing set, they design a model trained with proposed multifaceted cycle collaborative debiasing (MCCD) strategy that decreases the logit between any single modality and multimodality logit. MCCD also includes the loss function to maintain the similarity between bias logits. In experiments, they report results on both MUSIC-AVQA and MUSIC-AVQA-R testing sets. Comparing with previous methods, their proposed approach gets good performances in both MUSIC-AVQA and MUSIC-AVQA-R.

**Strengths:**

(1) The proposed MCCD strategy is model-agnostic. The experiments in Table 1 show the consistent improvements for previous methods when MCCD is applied on MUSIC-AVQA.
(2) This paper also contribute a new testing set for the research society to evaluate the robustness of AVQA models.
(3) The proposed model and MCCD achieves good performances in AVQA, obtaining comparable results on MUSIC-AVQA comparing with previous AVQA methods. It also outperforms previous methods in MUSIC-AVQA-R, showing the robustness of the proposed model.

**Weaknesses:**

(1) In ablation study, have the authors tried to only finetune the linear classifier using multimodal features? Without training with MCCD, I would like to see the performances on MUSIC-AVQA-R and MUSIC-AVQA.
(2) VisualBERT is originally trained with only image and text, but in the paper it seems VisualBERT is used to encode the audio information as well. Do the authors finetune VisualBERT during training?
(3) Even though the paper is easy to follow, the presentation can still be improved. In Figure 4, "Unimodal Bias Learning" has "Audio Bias Learner" but the input is actually video features. Some highlighted regions such as the purple region is not explained well.

**Questions:**

See weakness

**Limitations:**

yes

---

> ### Author Rebuttal · Authors · 2024-08-01
>
> Thanks for your valuable and constructive comments.
>
> * **Reply for Q1.** We fine-tune the VisualBERT and linear classifier simultaneously in the ablation study, consistent with the main experiment. The experimental results without training MCCD are shown in the following table. We observe that VisualBERT achieves competitive results on two datasets. This demonstrates that pre-trained or large multimodal models are more robust than smaller models, which aligns with the findings in [1] and [2]. We will add this experimental result and analysis to the revised manuscript.
>
> |           Method          | MUSIC-AVQA |           |        |        | MUSIC-AVQA-R |           |        |        |        |        |
> |:-------------------------:|:----------:|:---------:|:------:|:------:|:------------:|:---------:|:------:|:------:|:------:|:------:|
> |                           |  AQA  | VQA |  AVQA  |   All  |   AQA   | VQA |  AVQA  |    H   |    T   |   All  |
> | Ours              | 79.14   | 78.24   | 67.13    | 72.20   | 74.76   | **72.76**   | **61.34**   | **72.24**   | **59.39**   | **66.95**   |
> | w/o $d_i^{\mathrm{q}}$  | **79.64**   | 77.62   | **67.52**    | 72.34   | 74.89   | 72.26   | 59.53    | 71.72   | 57.47   | 65.86   |
> | w/o $d_i^{\mathrm{v}}$  | 79.27   | 78.61   | 67.23    | 72.37   | 71.29   | 70.18   | 58.08    | 68.34   | 57.43   | 63.85   |
> | w/o $d_i^{\mathrm{a}}$  | 77.22   | 77.50   | 67.31    | 71.76   | 70.92   | 67.82   | 55.57    | 66.05   | 55.61   | 61.75   |
> | w/o MD            | 78.46   | 77.54   | 66.39    | 71.48   | 73.97   | 70.41   | 58.87    | 70.09   | 57.25   | 64.80   |
> | w/o CG            | 78.77   | 78.65   | 67.50   | **72.45**   | **75.42**   | 71.72   | 59.68    | 71.40   | 57.98   | 65.87   |
> | w/o MCCD          | 77.85   | **78.81**   | 66.43    | 71.73   | 67.36   | 65.46   | 55.12    | 63.23   | 54.54   | 60.22   |
>
>
> [1] Wen, Z., Xu, G., Tan, M., Wu, Q., & Wu, Q. (2021). Debiased visual question answering from feature and sample perspectives. Advances in Neural Information Processing Systems, 34, 3784-3796.
>
> [2] Ma, J., Wang, P., Kong, D., Wang, Z., Liu, J., Pei, H., & Zhao, J. (2024). Robust visual question answering: Datasets, methods, and future challenges. IEEE Transactions on Pattern Analysis and Machine Intelligence.
>
> * **Reply for Q2.** We finetune VisualBERT during training. In Figure 1, we first employ ResNet-18 and VGGish to obtain 512-dimensional and 128-dimensional embeddings of the video and audio segments, respectively. Then, we utilize linear layers to ensure dimension matching for the input embeddings. Subsequently, we concatenate the video and audio embeddings to form context representations which are then input into the "visual_embeds" of VisualBERT.
>
> * **Reply for Q3.** We have re-painted this figure shown in Figure 1 of "response.pdf". We replace the purple region with a dashed box. The video and audio embeddings are concatenated to form context representations in multimodal learning, while they are input into VisualBERT to learn the single modality representation in uni-modal learning, respectively. This updated figure will be included in the revised manuscript. Thank you again for your suggestions.

---

> > ### Comment · Reviewer_Zk4k · 2024-08-13
> > **thank you**
> >
> > My concern is resolved and I just raised the score.

---

> > > ### Author Response · Authors · 2024-08-13
> > > **Letter of thanks**
> > >
> > > Dear reviewer,
> > >
> > > Thank you for improving the rating.
> > >
> > > Best Regards,

---

### Author Rebuttal · Authors · 2024-08-05

* **Reply for dataset quality and bias difference between AVQA and VQA.** **First**, we believe the rephrasing quality of MUSIC-AVQA-R can be guaranteed from three aspects: (1) professional annotators, (2) a strict screening mechanism, and (3) strong support from statistical indicators. For example, as mentioned in Section 3.1 and Appendix B, the Fleiss Kappa value used to measure consistency is 0.839, demonstrating high rephrasing quality. **Secondly**, we think the suggestion for comparing (train vs. test_head) and (train vs. test_tail) to be very valuable. The results, shown in "response.pdf," provide insightful analysis. In comparing the training and head splits, we observe that the distributions are similar, except for the "comparative" and "existential" questions. However, when comparing the training and tail splits, the distributions differ significantly. Therefore, performance on the head and tail splits can accurately reflect model robustness in both in- and out-of-distribution scenarios. We will also include this comparison result in the revised version of the manuscript. **Next**, we thank you for highlighting the three debiasing methods for VQA. Specifically, **[1]** generates various counterfactual training samples by masking critical objects in images or words in questions, assigning different ground-truth answers, thereby forcing VQA models to focus on all critical objects and words. **[2]** employs a greedy gradient ensemble strategy to eliminate language biases, including distribution and shortcut biases. Similarly, **[3]** uses a generative adversarial network and knowledge distillation loss to capture both dataset distribution bias and target model bias. To verify the effectiveness of these methods, integrations with non-debiasing VQA methods like UpDn [4] and debiasing methods such as RUBi [5] are conducted. However, we note that these methods are primarily designed for VQA, specifically addressing language bias, rather than for AVQA. The training bias in AVQA contains language, audio, and video bias. Taking STG [6] as an example, the experimental results on the MUSIC-AVQA test set are as follows. We see that this method can achieve 54% overall accuracy only depending on question inputs. It can be seen that relying solely on uni-modal bias can achieve good results. In addition, we can see that only depending on two modalities inputs obtain significant improvement, although AVQA is a task that involves three modalities. These results show that the mentioned training bias plays a key role in predicting answers.  **Therefore, we believe our method should be compared with both non-debiasing and debiasing AVQA methods.** Unfortunately, research on AVQA is still in its infancy, **with COCA being the only existing bias elimination method.** For this reason, our comparison is limited to non-debiasing AVQA methods. Following COCA [7], we compare our method with audio, visual, video, and AVQA baselines. Due to the lack of public code for COCA, we only compare our method with COCA on the MUSIC-AVQA dataset. Additionally, we think MCCD is specifically designed to mitigate three biases in the AVQA task. This strategy captures the audio, video, and language biases respectively, and then reduces the biases by enlarging the distance between the uni-modal logit and fusion logit, constrained by cycle guidance. To verify MCCD's effectiveness and plug-and-play capability, we conduct extensive experiments on two public datasets. We will discuss the aforementioned three works in the related work section.

| Input Modality | A Question | V Question | AV Question | All |
|:--------:|:-----:|:----------:|:-----:|:----------:|
| Q | 65.19 | 44.42 | 55.15 | 54.09 |
| A+Q | 67.78 | 62.75 | 63.86 | 64.26 |
| V+Q | 68.76 | 67.28 | 63.23 | 65.28 |

* **Reply for theoretical analysis of Equation (1).** **First**,  we focus on the relationship between distance and bias. In the context of logit magnitudes being constant, an increase in the distance between the uni-modal logit and the fusion logit generally implies a greater difference in their respective probability distributions. For example, in Figure 2 of "response.pdf", the distance between $A$ and $B'$ is larger than the distance between $A$ and $B$, which also makes the distribution difference (i.e. KL divergence) greater. Generally, considering the fusion logit distribution $X$ and a uni-modal logit distribution $Y$, the KL divergence $KL(Y||X)$ tends to increase as the distance between the logits grows. Furthermore, if we consider the mutual information $ I(Y;X) = H(X) + H(Y) - H(X,Y) $ between $X$ and $Y$, and utilize the definition of KL divergence, we get $ I(Y;X) = H(X) - KL(Y||X) $. This numerically indicates that, while keeping the fusion logit distribution fixed, a distance increase results in a mutual information decrease. This leads to a reduction in correlation between the uni-modal and fusion logit. **Several works [8], and [9] have discussed the relationship between mutual information and debiasing, supporting this observation.** **Next**, we will address why we opted to use the reciprocal of distance in MCCD. Our optimization goal is to maximize the separation between the fusion logit and uni-modal logit. Using $\frac{1}{d}$ as a metric allows for heightened sensitivity to small distance variations when the distance is small. Conversely, when a significant distance is established, further variations in distance have a relatively diminished impact. By employing $\frac{1}{d}$, the loss function becomes more effective in encouraging greater separation when logits are close to each other, thus ensuring that small distances are penalized more heavily. Additionally, $-d$ may be unsuitable: (1) may not handle the mentioned scenario, (2) fails to ensure nonnegativity of the total loss, and (3) may reduce the total loss and gradient. We hope this explanation clarifies the rationale behind MCCD and adequately addresses your concerns.

---

### Decision · Program_Chairs · 2024-09-25

**Decision:**

Accept (poster)

**Comment:**

This paper investigates bias in Audio-Visual Question Answering, introducing a new dataset and a method to mitigate the bias. All questions raised were addressed during the post-rebuttal discussion, leading to a consensus in favor of accepting the submission. The AC finds no significant concerns and agrees with this assessment.